# The daughter-parent plot: a tool for analyzing thermochronological data

Birk Härtel and Eva Enkelmann

Department of Earth, Energy and Environment, University of Calgary, Calgary, T2L 1N4, Canada

*Correspondence to*: Birk Härtel (birk.haertel@ucalgary.ca)

**Abstract.** Data plots of daughter against parent concentration (D-P plots) are a potential tool for analyzing low-temperature thermochronology, similar to isochron plots in radioisotopic geochronology. Their purposes are to visualize the main term of the radiometric age equation – the daughter-parent ratio – and to inspect the daughter-parent relationship for anomalies indicating influences of geological processes or analytical bias. The main advantages of the D-P plot over other data-analysis tools are: (1) its ability to detect systematic offsets in D and P concentrations, (2) its unambiguous representation of radiation damage dependent daughter retention, and (3) the possibility to analyze potential age outliers.

Despite these benefits, the D-P plot is currently not used for analyzing low-temperature thermochronology data, e.g. from fission track, (U-Th)/He or zircon Raman dating. We present a simple, decision-tree-based classification for daughter-parent relationships based on the D-P plot that places a dataset into one of seven classes: linear relationship with zero intercept, cluster, linear relationship with systematic offset, non-linear relationship, several age populations, scattered data, and inverse relationship. Assigning a class to a dataset enables to choose further data-analysis steps and how to report a sample age, e.g. as pooled, central or isochron age, or a range of ages. This classification scheme aims at facilitating thermochronological data analysis and making decisions more transparent. We demonstrate the proposed procedure by analyzing published datasets from a variety of geological settings and thermochronometers and introduce Incaplot, a graphical-user-interface software, that we developed to facilitate D-P plotting of thermochronology data.

## 1 Introduction

The isochron plot is a universal tool for analyzing geochronological results, e.g., U-Pb, Ar-Ar or Rb-Sr data (e.g., Nicolaysen, 1961). The main reason for its use is that the ratio of the isotope ratios (e.g., $^{87}Sr/^{86}Sr$ vs. $^{87}Rb/^{86}Sr$) on the plot's axes is the essential term of the radiometric age equation. The slope and intercept of an isochron fitted to a dataset convey information about the age and initial isotopic composition of a sample. Furthermore, the isochron plot enables us to visualize anomalous features in the data, such as outliers or excess of radiogenic daughters.

The isochron plot's equivalent for low-temperature thermochronology is the radiogenic daughter (D) vs. radioactive parent (P) plot (D-P plot), which several authors suggest for analyzing fission-track (FT), (U-Th)/He (He), and zircon Raman (ZR) data (e.g., Fanale and Kulp, 1962; Green, 1981; Wernicke and Lippolt, 1993; Dunkl, 2002; Vermeesch, 2008; He et al., 2021; Härtel et al., 2022a). This plot allows to: (1) detect systematic offsets in daughter or parent concentration (e.g., Vermeesch, 2008); (2) analyze the influence of radiation-damage on daughter retention avoiding spurious associations in the age-eU plot (Härtel et al., 2022a); and (3) evaluate single-grain ages in terms of a two-dimensional distribution (e.g., for detecting outliers), or selecting a sample age (e.g., as a mean, pooled, central, or isochron age) without a preconceived idea about the single-grain ages.

The D-P plot thus occupies the interface between the analytical results and more specific data-analysis tools such as radial, kernel-density-estimate (KDE), or age-grain size plots – a tool that helps us to decide which data-analysis techniques are applicable or not to a given dataset. It is therefore surprising that the D-P plot is not considered a standard tool for analyzing

thermochronological data (e.g., Flowers et al., 2022; Kohn et al., 2024). Our aim is to fill this gap and provide guidance to users of low-temperature thermochronology.

This article consists of two major parts: we first provide theoretical background of the D-P plot, its differences to the classic isochron plot, and give examples of commonly observed daughter-parent relationships (section 2). Then, we illustrate how to analyze data in D-P space, using a workflow for classifying daughter-parent relationships, suggest further data-analysis tools for each type of relationship and, if applicable, provide algorithms for sample age calculation (section 3). We also introduce Incaplot, a free graphical-user-interface software dedicated to create D-P plots that allows an easy implementation of our proposed analysis to FT, He, and ZR data.

## 2 Background

### 2.1 Deriving the D-P plot

Using a plot of daughter (D) against parent (P) concentration rests upon the general age equation:

$$t = \frac{1}{\lambda} \ln\left(1 + c\,\frac{D}{P}\right) \qquad (1),$$

where t is the age, λ is the decay constant, and c is a constant to balance out the units of D and P. It is evident from (1) that the age has a one-to-one relationship with D/P. Therefore, the position of a data point in a plot of D vs. P indicates the single-grain age by the slope of a line connecting it to the origin of the plot (Figure 1a). Data pairs from same-age grains plot on a line through the origin, representing a proportional relationship (Fig. 1b). The D-P plot is thus a graphical representation of the age equation.

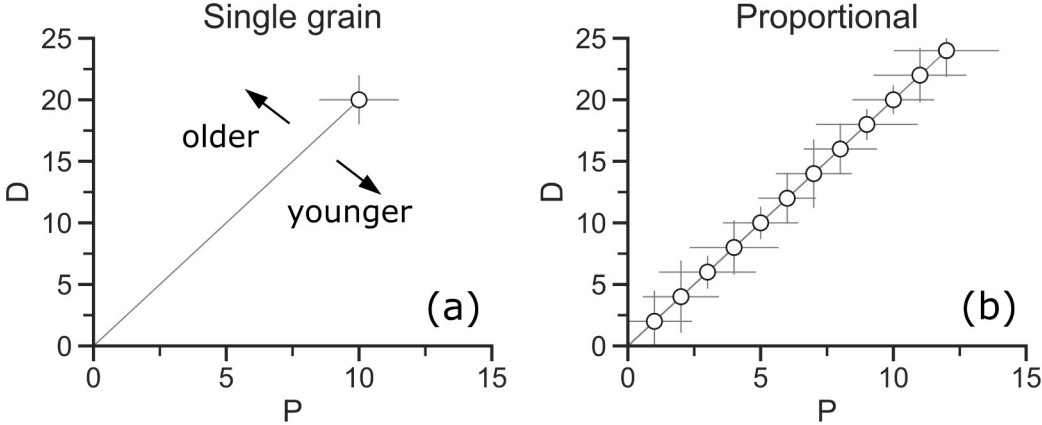

**Figure 1. Synthetic data displaying the basic concept of the D-P plot. (a) Single-grain datum in D-P space. The slope of the grey line connecting it to the origin corresponds to its D/P ratio. (b) Synthetic data with a constant D-P ratio forming an ideally proportional array.**

This relationship of the D-P plot to the age equation is the same as that of the classic isochron plot, but there are two significant differences: (1) the isochron plot represents parent and daughter concentrations as isotope ratios with a non-radiogenic sister isotope as the common denominator. This creates error correlation between the two axes of the plot, which is not present in the D-P plot as it relies on independently measured daughters and parents. (2) The isochron plot assumes the initial presence of the radiogenic daughter isotope, which makes isochron fitting indispensable for age calculation. In contrast, for the D-P plot no initial daughters are assumed, enabling the analyst to examine the D-P relationship for patterns without the need for an isochron. To honour these differences, we prefer the generic term *D-P plot* over *isochron plot* for this type of diagram for FT, He, and ZR data.

The actual quantities of D and P depend on the dating method. For FT dating, the daughters are the number or areal density of spontaneous tracks and the parents are either that of induced tracks (external detector method) or U concentration (LA-

ICP-MS-based dating). The daughters for He and ZR dating are the α-ejection corrected He concentration and the radiation-damage density, respectively. However, defining a parent concentration for these methods is difficult, because several α-emitting nuclides – $^{238}U$, $^{235}U$, $^{232}Th$ (and $^{147}Sm$) – have to be considered. One solution is to express the parents as an effective uranium concentration (eU) – the sum of the parent concentrations weighted by their relative α-production rate. This reduces the number of parents to one. Appendix A discusses the calculation of eU as a parent concentration in (U-Th)/He and zircon Raman dating and the differences between existing eU equations (e.g., Cooperdock et al., 2019; Härtel et al., 2023). Appendix B provides additional discussion on the choice of daughter and parent concentration units for different dating methods.

## 2.2 Data patterns for multi-grain samples

In practice, the analyst acquires multiple single-grain data to extract information about a sample's thermal history. The number of these single-grain analyses varies between methods and analytical protocol – from about 20–30 grains per sample for FT and ZR dating to only 3–5 grains per sample for whole-grain He dating. The D-P plot allows us to analyze such multi-grain samples. However, real data deviate from the ideal trend in Fig. 1b. In the following, we give a short overview of the typical deviations from the ideal proportional D-P relationship and the information they contain regarding geological processes that influence rock cooling and heating or analytical biases.

### 2.2.1 Linear relationship with zero intercept

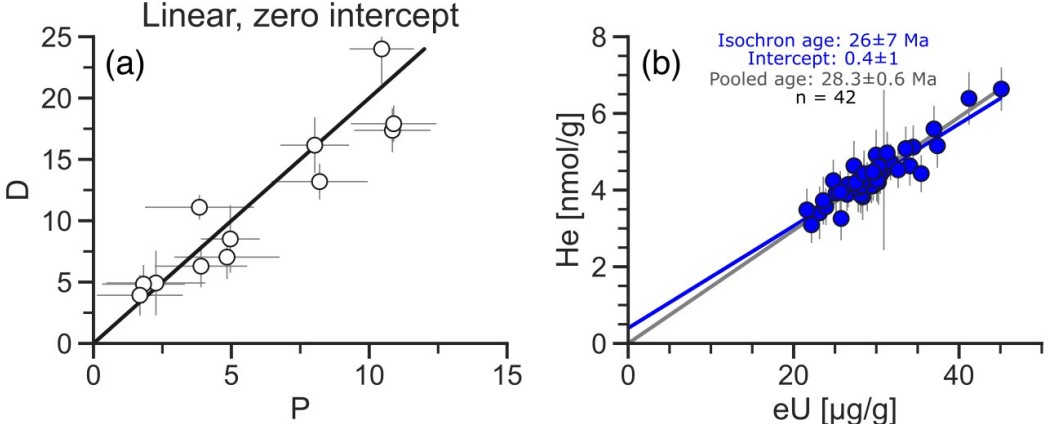

Figure 2. Linear D-P relationships with zero intercept. (a) D-P plot showing a linear relationship with zero intercept in synthetic data. (b) D-P plot of laser-ablation apatite He data showing a linear trend with an (approximately) zero intercept (Fish Canyon Tuff apatite, Pickering et al., 2020). The blue line is a robust isochron; the grey line connects the origin and the mean D and P values representing the pooled age. The uncertainties in b are 2s.

Fig. 2a presents a synthetic example of a positive linear D-P relationship with a zero intercept, including random variation about the trend. This is similar to the proportional case, but with uncertainty on the D and P measurements. Additional variation may be the consequence of varying grain sizes or inaccurate α-ejection correction for He dating, inter-grain chemical differences for FT or ZR dating, and parent-concentration-zoning for all three methods. The D-P plot in Figure 2b shows an example of a linear relationship with a zero intercept for laser-ablation apatite He data from the Fish Canyon Tuff reference material (Pickering et al., 2020). Both, the isochron (26 ± 7 Ma) and the pooled age (28.3 ± 0.6 Ma) overlap with the reference age at 28.4 Ma (Schmitz and Bowring, 2001)

### 2.2.2 Cluster

Fig. 3a shows a synthetic example of clustered D-P data. This pattern is typical for data from samples with limited inter-grain differences in parent (and daughter) concentrations, and usually their uncertainty intervals overlap strongly. In this

case, the positive relationship between daughters and parents may be obscured by the uncertainty. Figure 3b shows an example of a D-P plot with clustered apatite FT data from sample FC-1 from the Duluth Complex, Minnesota (Härtel et al., 2022a). Despite relatively large differences in track density, the uncertainties in D and P of most grains overlap. The data give a pooled age of 850 ± 30 Ma.

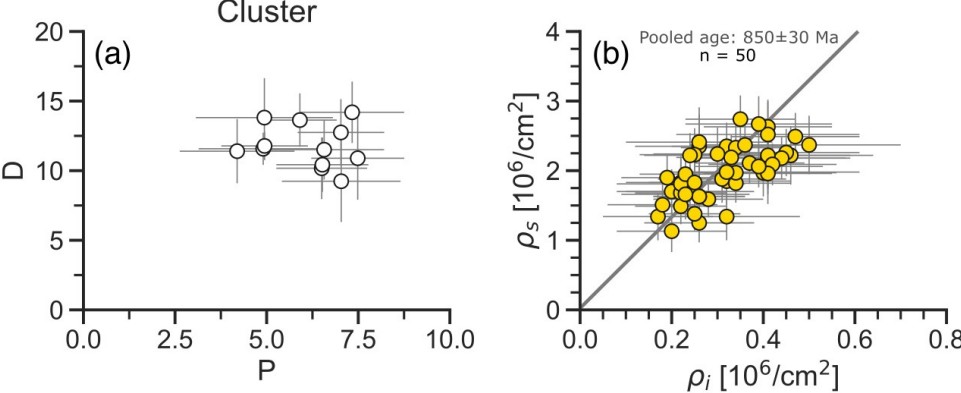

**Figure 3. Clustered D-P relationships. (a) D-P plot showing clustered synthetic data. (b) -P plot of apatite fission-track data forming a cluster (FC-1 apatite, Härtel et al., 2022a). The grey line is a tangent through the origin and the mean D and P values representing the pooled age. The uncertainties in b are 2s.**

### 2.2.3 Linear relationship with systematic offset

In Fig. 4a, the synthetic data form a linear trend, which, compared to Figs. 2a and 2b, is offset from the origin. In He dating, such an offset may result from (1) 'parentless helium' implanted by inclusions (Vermeesch et al., 2007), eU-bearing grain-boundary or neighboring phases (e.g., Spiegel et al., 2009; Murray et al., 2014) or (2) a consistent style of zoning across grains affecting α-ejection correction (e.g., Orme et al., 2015). In FT dating, it may also be due to a bias towards higher or lower track counts (see Green, 1981). In ZR dating, systematic offsets may result from damage-calibration issues, asymmetric Raman bands, or composition-related Raman-band broadening (Kempe et al., 2018; Troch et al., 2018; Härtel et al., 2022b). Note that an over- or underestimation of P causes an apparent offset of opposite sign in D.

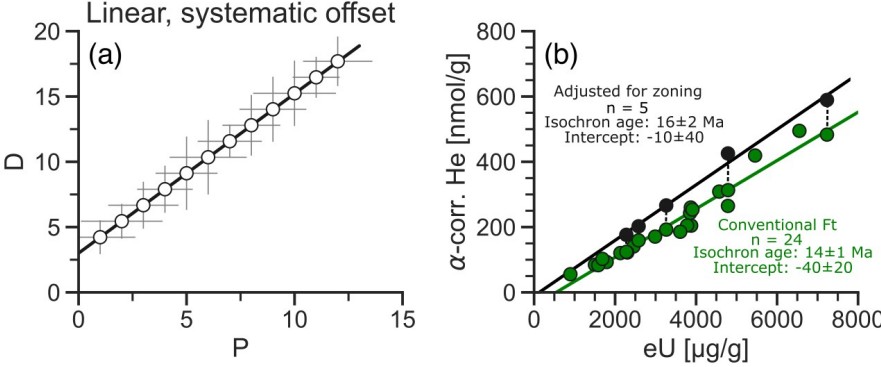

**Figure 4. Linear D-P relationships with systematic offset. (a) D-P plot showing a linear relationship with systematic offset in synthetic data. (b) D-P plot of whole-grain zircon He data forming a linear trend with a negative offset (green) and data points from the same grains with adjustment for zoning (black) with isochrons (calculated from least-squares regression; data from Miocene leucogranite, Orme et al., 2015). Dashed black lines connect adjusted data to their unadjusted counterparts.**

The D-P plot in Figure 4b shows an example of negative offset in whole-grain zircon He data from a set of four closely spaced samples of Miocene leucogranite from the Greater Himalaya sequence (Orme et al., 2015). The single-grain ages range from 9.9–14.7 Ma (weighted means: 10–12 Ma), whereas Orme et al. (2015) expected an age range of 14–17 Ma due to host-rock stratigraphy and other thermochronological data. They explained this by the zircon grains consistently showing

compositional zoning with low-eU cores and high-eU rims: this causes more He to be lost by α-ejection than accounted for by conventional Ft-correction (e.g., Hourigan et al., 2005) and leads to the negative offset. They tested this assumption by adjusting Ft of some grains using zoning information from laser-ablation depth drilling (black circles in Fig. 4b). The ages range from 14.8 to 17.0 Ma (weighted mean: 15.6 ± 0.2 Ma). In the D-P plot, these data points fall above the main trend and show insignificant offset from the origin. The isochron ages for both, unadjusted (14 ± 1 Ma) and adjusted data (16 ± 2 Ma) overlap with each other and with the expected age range.

### 2.2.4 Non-linear relationship

Figure 5a showcases a synthetic example of a non-linear D-P relationship. This may be due to the daughter retention depending on the degree of lattice damage from α-decay of U, Th, and their daughters. The production of radiation-damage is roughly proportional to the parent (eU) concentration. Its effect on daughter retention causes D and P to form either a concave (Fig. 5a, damage-enhanced loss) or a convex (damage-enhanced retention) relationship (Härtel et al., 2022a).

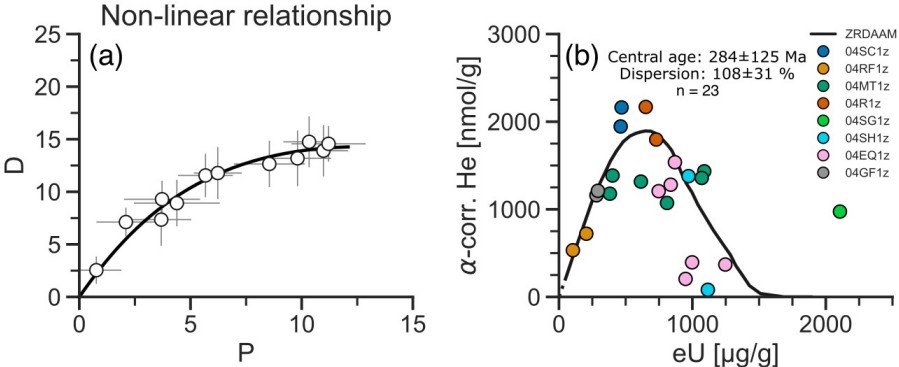

Figure 5. Non-linear D-P relationships. (a) D-P plot showing synthetic data with a non-linear relationship. (b) D-P plot of whole-grain zircon He data showing a non-linear, concave relationship (Minnesota River Valley samples from Miltich, 2005). The line represents the predicted D-P trend of a zircon radiation-damage and annealing model (ZRDAAM) from Guenthner et al. (2013). The dotted line segment on the left connects the ZRDAAM estimate with the origin.

Figure 5b shows an example for a non-linear D-P relationship due to radiation-damage-enhanced helium loss in zircon He data from the Minnesota River Valley (Miltich, 2005). The dataset consists of several samples assumed to have shared the same thermal history since ~1.8 Ga based on earlier thermochronological data (see references in Miltich, 2005). The He concentration increases approximately linearly with eU increasing up to 500 μg/g and falls at higher eU concentrations in response to radiation damage facilitating He loss from the zircon crystals. Guenthner et al. (2013) suggested a thermal history for these samples based on the zircon radiation-damage accumulation and annealing model (ZRDAAM, black line), consistent with the D-P relationship.

### 2.2.5 Several populations

The synthetic example in Fig. 6a shows data forming two linear trends in the D-P plot, indicating different age components within the sample. Such a trend occurs if a sample contains groups of grains with a high contrast in kinetic properties. Figure 6b shows a D-P plot for an example of different age populations found in apatite FT data for a fully reset sedimentary sample from the Mackenzie Basin, Northwest Territory (sample I-77; Issler et al., 2005). It displays two roughly linear trends in the data corresponding to two different ages (90 ± 12 Ma and 220 ± 45 Ma). Color-coding the data by the chlorine content shows a slight compositional difference between the two age populations, suggesting a chemical influence on FT annealing properties (e.g., Barbarand et al., 2003).

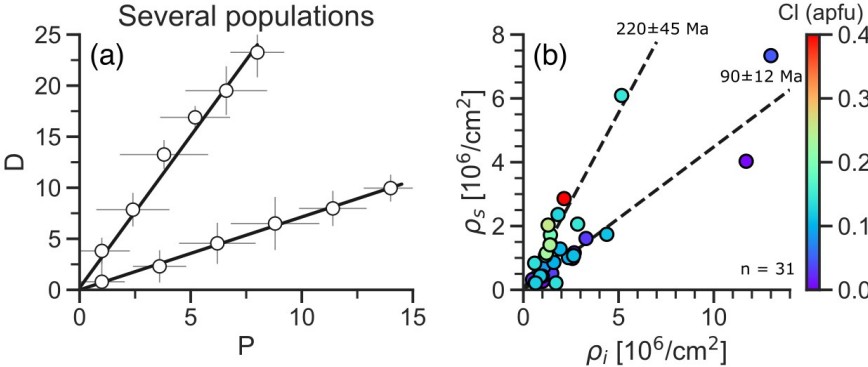

Figure 6. Several-population D-P relationships. (a) D-P plot showing synthetic data with two populatoins. (b) D-P plot of apatite fission-track data showing two populations (sample I-77, Issler et al., 2005). The data are color-coded by chlorine content (in atoms per formula-unit for $Ca_{10}(PO_4)_6(F, OH, Cl)_2$). The dashed lines represent ages determined from finite-mixture modelling (Issler et al., 2005).

### 2.2.6 Scattered data

The D-P plot of synthetic data in Fig. 7a shows how random scatter can obscure the relationship of D and P. Such a pattern may result from different factors, e.g., heterogeneous daughter retention within the sample, e.g. a broad range of grain sizes or chemical compositions. Other reasons for scattered data might be the occurrence of micro-cracks, deformation, or parent zoning. In addition, scatter may arise from analytical factors, such as variably biased α-ejection correction, counting bias, or a combination of these factors.

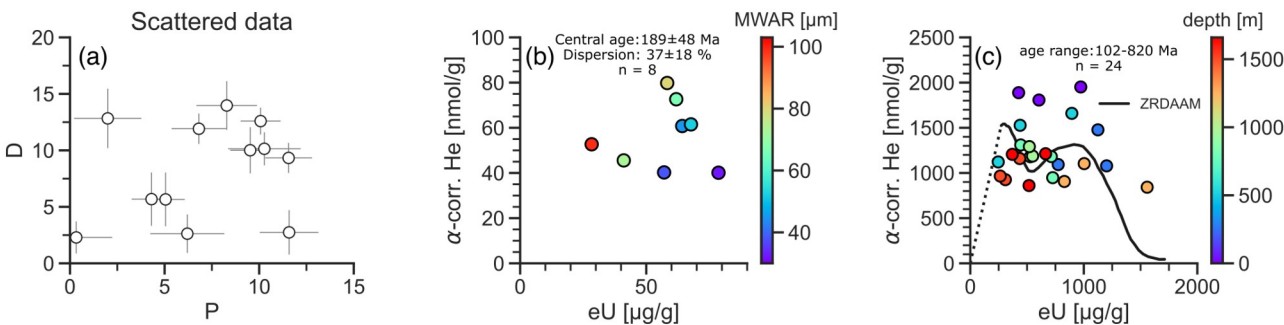

Figure 7. Scattered D-P relationships. (a) D-P plot showing randomly scattered synthetic data. (b) D-P plot of multi-grain-aliquot apatite He data showing a scattered relationship (Shell Canyon 12 sample from Reiners and Farley, 2001). The data are color-coded by grain size expressed as mass-weighted average radius (MWAR). (c) D-P plot of whole-grain zircon He data showing a scattered relationship (Laxemar region samples from Guenthner et al., 2017). The line represents the predicted D-P trend from ZRDAAM, and the dotted line segment on the left connects to the origin.

Figure 7b shows an example of scattered data in the D-P plot from the multi-grain-aliquot apatite He data from the Bighorn Mountains, Wyoming (Shell Canyon 12; Reiners and Farley, 2001). The data show no relationship between He and eU. However, color-coding the different aliquots by the mass-weighted average radius (MWAR) reveals an increasing age (i.e., D/P ratio) with increasing grain size. This indicates a continuous age distribution due to different sensitivity of differently sized grains to volume diffusion of helium. Figure 7c shows another example of a scattered D-P relationship in whole-grain

zircon He data from the Laxemar region on the Fennoscandian Shield (Guenthner et al., 2017). The color-coding reflects the sampling depth and the black line represents the ZRDAAM from the original publication. However, neither the depth of each sample – a proxy for their current temperature – nor the radiation-damage model explain the scatter in the data. In this case, an unknown factor causes the age variation.

**2.2.7 Inverse relationship**

Figure 8a shows a synthetic dataset with an inverse relationship in the D-P plot. This pattern may occur due to (1) a small sample size causing a spurious relationship (Ketcham et al., 2018), (2) bias from over- or under-correcting the He concentration for α-ejection, or (3) the data representing a falling segment of a non-linear trend caused by radiation damage. Figure 8b provides an example for a negative D-P trend from whole-grain zircon He data from Proterozoic samples from Baffin Island (sample A10-42, Ault et al., 2018; Armstrong et al., 2024). Ault et al. (2018) interpreted the age variation in

this dataset as due to radiation-damage enhanced He loss, as the ZRDAAM (black line) in Figure 8b shows. Armstrong et al. (2024) provided additional Raman data on selected grains, showing that some of the zircon grains with eU ≥ 1000 µg/g were metamict, pointing to enhanced He loss compared to the lower-eU grains (Guenthner et al., 2013).

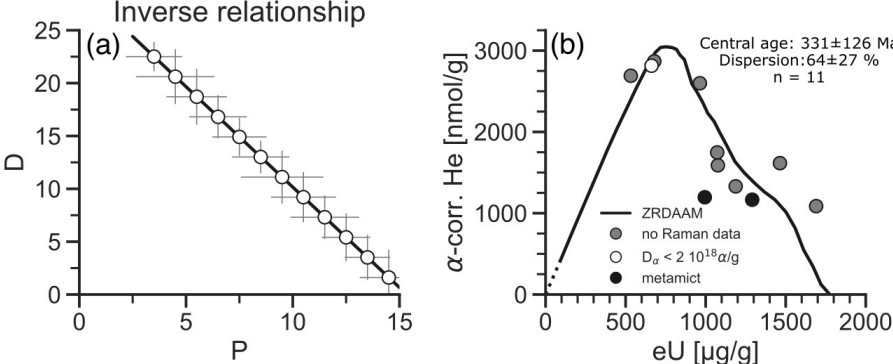

**Figure 8. Inverse D-P relationhips. (a) D-P plot showing inverse relationship in synthetic data. (b) D-P plot of whole-grain zircon He data showing an inverse relationship (sample A10-42 from Ault et al., 2018, and Armstrong et al., 2024). Color-coding indicates radiation-damage measurements using a Raman microprobe. The line represents the predicted D-P trend from ZRDAAM, and the dotted line segment on the left connects to the origin.**

**2.3 Unique benefits of D-P plots**

Figures 2-8 show the range of D-P patterns that occur in thermochronological data. While the mean D/P ratio of the synthetic

datasets shown in these figures is the same (2), each of these relationships requires different considerations for data analysis. This includes the questions if reporting a single sample age is appropriate, and if yes, which type of sample age to report. Commonly used data-analysis tools such radial plots, KDE, or age-grainsize plots help to trace some of the factors causing age variations, but there are unique benefits to analyzing data in the D-P plot.

First, the D-P plot is the only thermochronological data plot that enables us to *detect systematic offset* in daughter or parent

concentrations (Figs. 4a, b). Systematically offset data pose a serious problem to many standard data analysis tools and should therefore be treated with caution: (1) single-grain ages calculated from offset data are biased towards higher or lower ages depending on the sign of the offset. This bias propagates into calculated central tendencies (Härtel et al., 2022a) and into plots displaying the age as a variable, such as radial, KDE, age-grainsize, and age-(e)U plots. (2) Offset data appear over-dispersed (and fail the $\chi^2$ test) because the data uncertainties do not explain the spread in age. This further complicates

the use of radial plots, as the spread in single-grain ages may give way to a misinterpretation of ages as a mixture of discrete age components (see discussion in Vermeesch, 2019). (3) The over-dispersion by systematic offset hampers inverse thermal-history modeling, as the modeling algorithm will have to reconcile a large spread in ages without the uncertainties accounting for it (e.g., Vermeesch and Tian, 2014). The offset affects each data point differently, so that expanding the uncertainties in D and P does not solve this problem (see Flowers et al., 2022). (4) Systematic offset also compromises the

Helioplot (Vermeesch, 2010), which determines the age from log-ratios, because it disturbs all ratios derived from the D and P concentrations.

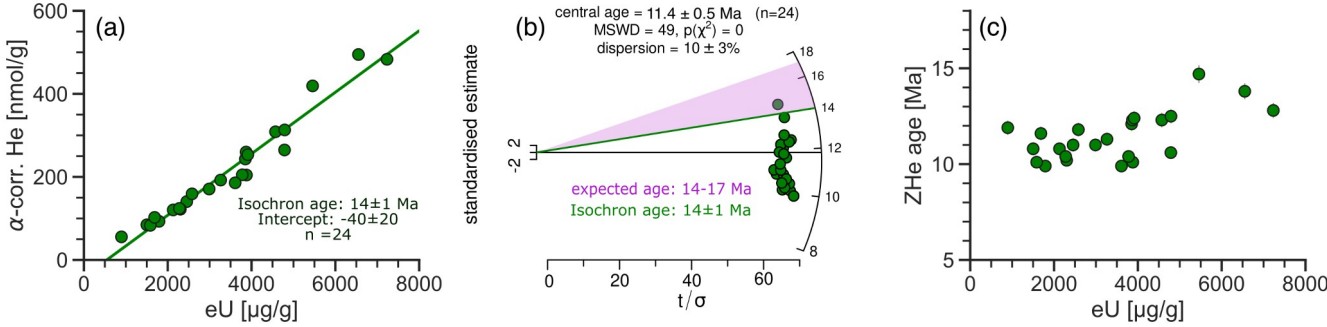

**Figure 09. Example for the bias caused by systematic offsets. (a) Radial plot of the zircon He data from Orme et al. (2015) shown in Figure 4b. The center of the y-axis is the central age. The purple area represents the expected age range from stratigraphic and thermochronological constraints and the green line is the isochron age from (a). (c) Age-eU plot of the same data showing a weak positive association.**

Figure 9 illustrates some these biases in more traditional data-analysis tools using the zircon He data of Orme et al. (2015, Fig. 4b). Figure 9a reproduces the D-P plot of the data in Figure 4b showing the negative, systematic offset, which is the result of a biased α-ejection correction due to consistent compositional zoning (Orme et al., 2014). However, the radial plot in Fig. 9b does not show any anomaly in the data except for overdispersion (P($\chi^2$) ≈ 0; dispersion = 10 ± 3 %). The single-grain ages (9.9–14.7 Ma) and the central age (11.4 ± 0.5 Ma) are substantially younger than the age range of 14–17 Ma, established from stratigraphic and thermochronological constraints (purple area; Orme et al., 2015). In contrast, the isochron age (14±1 Ma) from the D-P plot fits well with this scenario. In the age-eU plot (Fig. 9c), the data show a weak, positive association. However, this association is misleading in comparison to the usual interpretation of associations in the age-eU plot (see below) because the age variation arises from biased α-ejection correction and not a radiation-damage effect on He retention (Guenthner et al., 2013; Gautheron et al., 2020).

Second, the D-P plot provides an unbiased indication if daughter retention in a sample depends on *radiation damage*: the D-P plot shows unambiguous non-linear or inverse relationships for well-documented cases of radiation-damage-dependent daughter retention (e.g., Figs. 5b, 8b). The commonly used age-eU plot also shows relationship for such cases (e.g., Miltich, 2005; Guenthner et al., 2013; Ault et al., 2018; Armstrong et al., 2024). However, not all associations observed in the age-eU plot reflect actual radiation-damage effects, but may be the result of spurious ratio correlation (e.g., Carter, 1990; Härtel et al., 2022a). Figure 10 illustrates this problem using the examples from Figs. 7 and 8. Figures 10a and d show the D-P and age-eU plots for the zircon He data from Fig. 8b with an inverse daughter-parent relationship due to radiation-damage-dependent He retention (color-coding and black line in Fig. 10a; Ault et al., 2018; Armstrong et al., 2024). The age-eU plot shows a negative association as expected for zircon with a high radiation-damage density (Guenthner et al., 2013). For comparison, Fig. 10b and e show the D-P and age-eU plot for the scattered zircon He data from Fig. 7c (Guenthner et al., 2017), for which a radiation-damage model does not explain the variation in daughters and parents (black line in Fig. 10b). The age-eU plot in Fig. 10e shows a similar negative association to Fig. 10d. Also, the apparent relationship between age and eU concentration masks the scatter clearly visible in Fig. 10b. Figure 10c and f show the D-P and age-eU plot for the scattered apatite He data from Fig. 7b (Reiners and Farley, 2001), whose variation is explainable by grain-size differences. However, the age-eU plot in Fig. 10f also shows a negative association. In this case, the effect of grain size on He diffusion causes daughter and parent concentrations to vary and this variation translating to a spurious association when reprojecting the data in age-eU space. The D-P plot therefore reliably detects radiation-damage effects, whereas the age-eU plot often displays false-positive age-eU associations. Härtel et al. (2022a) provide more examples and discussion on spurious age-eU associations.

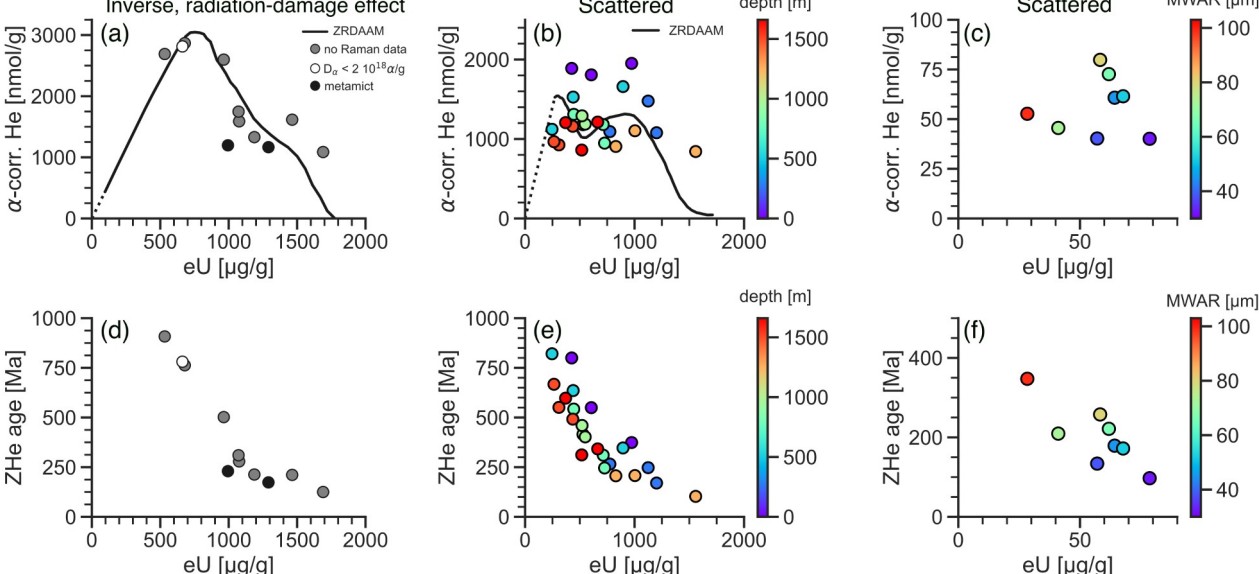

**Figure 10. Examples for detecting radiation-damage effects in age-eU and D-P plots. (a-c) D-P plots of data examples from Ault et al. (2018) and Armstrong et al. (2024, Fig. 8b), Guenthner (2017, Fig. 7c), and Reiners and Farley (2001, Fig. 7b), showing an inverse relationship due to radiation damage (a), a scattered relationship of unknown origin (b), and a scattered relationship due to grain-size differences (c). (d)-(f) Age-eU plots of the data in (a)-(c) showing negative associations.**

Third, the D-P plot allows to *detect age outliers* in two-dimensional space, not only from single-grain ages (e.g., He et al., 2021). It thus allows to identify the relative position of outliers with respect to the rest of the data, showing if its main deviation occurs in D or P. In addition, it is advantageous to identify outliers without directly considering the single-grain ages, as this may bias the detection in favor of grains that fit well with an *a priori* assumption of the sample age.

These advantages support a unique perspective of the D-P plot, which allows to side-step biases that other data-analysis tools show towards data with certain D-P relationships, most notably systematically offset data. We therefore suggest the D-P plot as a *first step* for thermochronological data analysis to identify the D-P relationship. Based on this relationship, it is possible to choose unbiased, more specific data-analysis tools such as radial, KDE, or age-grainsize plots, or thermal-history modelling. The following section presents a practical approach to using the D-P plot for data analysis.

## 3 Classification-based workflow for data analysis using the D-P plot

Our data-analysis scheme rests on a decision-tree approach to classify the daughter-parent relationship (Fig. 11). Depending on the class of the relationship, we then suggest further steps of data analysis. The following sections outline the use of the decision tree to systematically classify the data and find an appropriate description of the contained thermal-history information.

### 3.1 Preliminary considerations

Before using the classification scheme in Fig. 11, it is essential to assure that the analytical procedures and samples meet certain quality criteria established for each method, e.g., that suitable grains were selected for He dating, that data with asymmetric Raman bands were excluded from ZR dating, that track counting was conducted on prismatic grain surfaces, etc. Also, the number of analyses in the dataset is important, as fitting a regression line or splitting a dataset into age populations is not appropriate for small datasets (see sect. 3.5). Another criterion to be considered is the geological background of the sample. For example, a crystalline bedrock sample with a simple cooling history will likely give a single age, while a metasedimentary rock may show different age populations due to chemical variation between grains, and a volcanic rock recording its eruption is expected to give a near-ideal linear trend.

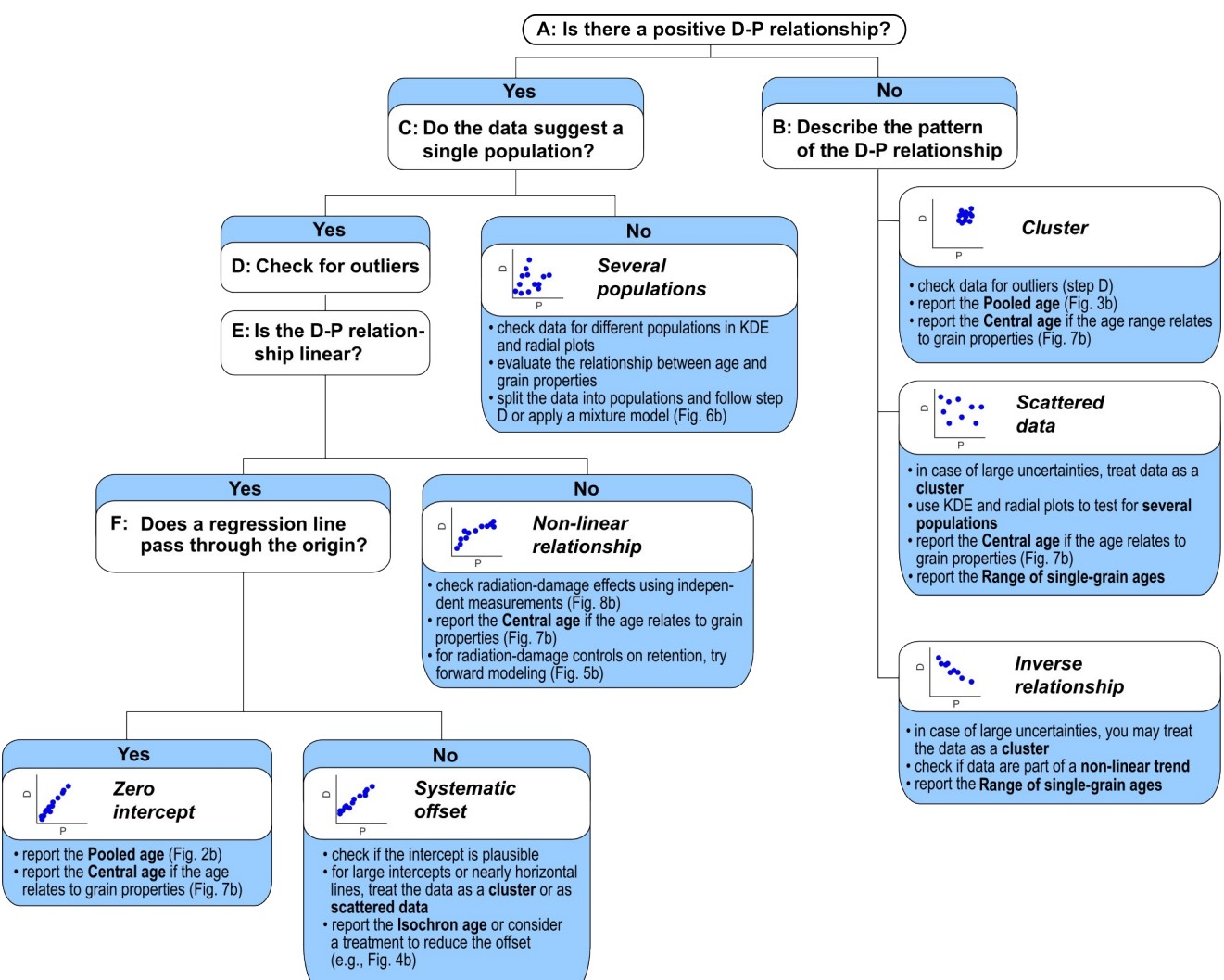

**Figure 11. Decision tree for classifying the daughter-parent relationship in a sample (classes named as in Figs. 2-8). The blue boxes provide suggestions on how to treat data belonging to the respective class.**

Radiation-damage effects and accompanying non-linear relationships are expected for old rocks with protracted or complex cooling histories, but not for young rocks that did not spend time in the temperature regime of radiation-damage accumulation. The interpretation of a sample that strongly deviates from the geological expectations needs to be carried out with care.

**3.2 The classification procedure**

For analyzing the data, we calculate the daughter and parent concentrations according to the thermochronological method used (see Appendix B) and plot daughters against parents. The analysis proceeds by following the decision tree in Fig. 11 to classify the daughter-parent relationship. The first step separates datasets showing a positive D-P relationship from those that do not (A in Fig. 11). We expect a positive association between D and P from the radioactive production equation, but this association may be obscured by factors discussed in sect. 2.2. In the case of data, for which the D-P relationship is not clear,

it is usually safe to assume that there is no positive relationship – a decision that may be revised in later steps. Data, for which D and P are not positively associated, are then classified as either clustered, scattered or following an inverse relationship (B in Fig. 11).

For data with a positive D-P relationship, it is then essential to distinguish datasets containing a single age population from those with several populations (C in Fig. 11). As in Fig. 6, multiple age populations form linear arrays with different slopes

or clusters in the dataset with gaps between them. A KDE plot may reveal the presence of different populations for cases that are not clear-cut.

For single-population data, the next step is filtering the dataset for outliers (D in Fig. 11). Outliers stick out by a difference in single-grain age to the other data beyond their uncertainty. However, this is not sufficient evidence to mark a data pair as anomalous: other factors such as systematic offset may also cause single grains to be significantly older or younger than the

250 others (Figs. 4b, 9b). In the D-P plot, outliers show up as removed from the main trend or group of data points. Before considering such a measurement as anomalous, other properties should be examined, e.g. grain size or mineral chemistry. If anomalous data are excluded from further analysis, this should be reported, e.g. by marking the excluded data point as empty symbol in the D-P plot. For ambiguous cases, it may be advantageous to carry out the further steps with and without the concerned data point. For He dating, Flowers et al. (2022) provide further strategies for treating outliers (their sect. 3.1).

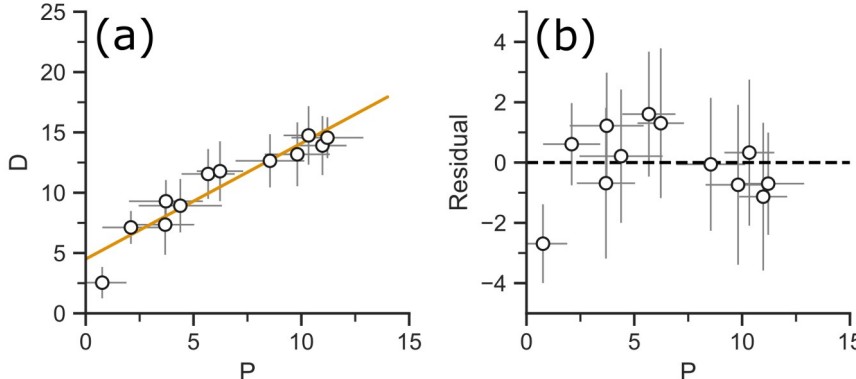

**Figure 12. D-P plots for testing for a non-linear relationship. (a) Regression line fit to a synthetic non-linear D-P relationship. (b) Fitting residuals (difference between measured and fitted D).**

After examining the outliers, we test the data for a linear D-P association (E in Fig. 11). If it is not clear whether the data show a linear or a non-linear trend from visual inspection alone, this can be verified by fitting a regression line to the data and examining the residuals, i.e. the deviation of the data points from the line. For a linear relationship, the residuals scatter randomly around zero while in the case of a non-linear relationship, there is an association between residual and parent

concentration. Fig. 12a shows the linear fit to a synthetic dataset in a D-P plot. Figure 12b plots the fitting residuals against P, revealing a boomerang-shaped trend that points to a non-linear D-P relationship.

If the D-P relationship is linear, it is necessary to test the data for a systematic offset (F in Fig. 11). This is achieved by fitting a regression line to the data and examining its intercept. If the intercept includes zero in its uncertainty envelope, the offset is not significant and the data may be treated as having a zero intercept. If the uncertainty envelope does not include

zero, this is a sign for a potential offset. However, this uncertainty on the intercept may be an underestimate if the variation of the data strongly exceeds that expected from the uncertainties (e.g., high MSWD; Wendt and Carl, 1991; see Appendix C). Another simple test for an intercept is the comparison of the isochron age and the pooled age: if the data form a trend through the origin, the two ages should be indistinguishable because the pooled age assumes a zero intercept (see sect. 3.4.1).

### 3.3 Sample-age calculation

Once arrived at a certain class of D-P relationships, the goal is to assign an age to the sample. This can either be a central tendency, such as a mean or pooled age, or an isochron age for a sample with a single age population, or a number of ages or a range of single-grain age depending on the D-P relationship. If the given ages can be described by a single sample age, the simplest solution is to report a central tendency. Despite its simplicity, the (arithmetic) mean age does usually not provide a

reliable sample age (e.g., Vermeesch, 2008; Härtel et al., 2022a; see Appendix C). A more robust alternative is the pooled age, which uses the ratio of the summed D and P concentrations (see Figs. 2b, 3b).

If the intra-sample age variation are related to a certain grain property affecting radiogenic daughter retention, the ages may represent a continuous mixture of ages, with each grain recording a different age due to its individual properties. Figure 7b shows an example with He data varying with respect to grain size. Such a mixture is best described by the central age (e.g. Galbraith, 2005; Vermeesch, 2019), or by thermal-history modeling taking into account the specific grain property.

Datasets, that are systematically offset, require a different approach, that of the isochron age, which rests on the slope of a fitted regression line through the D-P data (see Fig. 2b, 4b). If several discrete age components exist in a dataset, these can be separated by mixture modeling (e.g., Galbraith and Laslett, 1993; Vermeesch, 2019), or by treating each age component as a single sample (see Fig. 6). If the data cannot be described by a single age or multiple ages, nor by a continuous mixture related to grain properties, it is still possible to report the range of single-grain ages, which does not rely on any model assumptions. Appendix C provides a more detailed discussion about mean and isochron ages, and discrete and continuous age mixtures.

Table 1 shows an example for reporting format using the data from Figures 2–8. To make the process of data analysis transparent, we recommend to either show the daughter-parent plot for each sample or at least report the class of the D-P relationship and the type of the reported age. Table 1 shows an example for reporting format using the data from Figures 2–8. The following sections provide specific suggestions for how to treat data falling into each of the D-P classes of Fig. 11.

### 3.4 Classes of daughter-parent relationship

### 3.4.1 Linear relationship with zero intercept

If the daughter-parent relationship is linear and the intercept of its regression line is close to zero (F in Fig. 11), the pooled and the isochron age are similar (Fig. 2b). In this case, it is advantageous to report the pooled age, which is more robust and does not require the intercept as additional parameter. As all single-grain ages along the linear trend are roughly the same, the potential bias of the pooled age is negligible (see Appendix C). If the MSWD or spine factor of the fitted regression line (F in Fig. 11) are outside the upper confidence limit, the data are over-dispersed. This points to two possible scenarios: (1) Analytical dispersion due to the uncertainties not reflecting the actual measurement error. This is especially a problem for He and laser-ablation FT dating (e.g., Fitzgerald et al., 2006; Ketcham et al., 2018; Cogné and Gallagher, 2021). In this case, the uncertainty on the pooled age may be expanded to account for the variation of the individual analyses (see Eq. (C6) in Appendix C). (2) Geological dispersion due to heterogeneous grain properties affecting daughter retention, such as grain size, composition etc. This can be tested by plotting the age against these properties, or by using them for color-coding the D-P plot (Fig. 7b). If the data are dispersed due to a continuous range of grain properties, the central age describes the age distribution best (Appendix C). In this case, thermal-history modeling may take into account the variation of this grain property (e.g., grain size or radiation damage).

**Table 1. Example for reporting data-analysis results based on the D-P plots in Figs. 2–8.**

| Sample name(s) | Method | D-P relationship | Age reported | Age (Ma) | n | Comment | Reference | Figure |
|---|---|---|---|---|---|---|---|---|
| FCT | AHe (LA) | Linear, zero intercept | Pooled age | 28.3±0.6 | 42 | - | Pickering et al., 2020 | 2b |
| FC1 | AFT (EDM) | Cluster | Pooled age | 850±30 | 50 | - | Härtel et al., 2022a | 3b |
| Multiple samples, list in reference | ZHe (WG) | Linear, offset | Isochron age | 14±1 | 24 | Conventional Ft correction, Intercept: -40±23 | Orme et al., 2015 | 4b |
| Multiple samples, list in Fig. 3a | ZHe (WG) | Non-linear | Central age | 284±125 | 23 | Dispersion: 108±31 % Interpretation from radiation-damage model | Miltich, 2005 | 5b |
| I-77 | AFT (EDM) | Several populations | Finite mixture ages | 220±45 90±12 | 31 | Interpretation by retention depending on chlorine content | Issler et al., 2005 | 6b |
| Shell Canyon sample 12 | AHe (MG) | Scattered | Central age | 189±48 | 8 | Dispersion: 37±18 % Interpretation from grain-size model | Reiners and Farley, 2001 | 7b |
| Multiple samples, list in reference | ZHe (WG) | Scattered | Single-grain age range | 102-820 | 24 | - | Guenthner et al., 2017 | 7c |
| A10-42 | AHe (whole grain) | Inverse | Central age | 331±126 | 11 | Dispersion: 64±27 % Interpretation from radiation-damage model | Ault et al., 2018; Armstrong et al., 2024 | 8b |

Note: LA – laser-ablation, EDM – external-detector method, WG – whole-grain method, MG – multi-grain aliquot method, AHe – apatite He, AFT – apatite FT, ZHe – zircon He.

The age uncertainties are 2s.

### 3.4.2 Cluster

Clustered data are best summarized by the pooled age. It is advantageous to display the pooled age as a line in the D-P plot (Fig. 2b) to check if the single-grain D and P uncertainties overlap with the pooled age. To make sure that there is no bias towards the oldest or highest-D-P grains, the data should be screened for outliers (D in Fig. 11). If the data are over-dispersed, e.g., failing the $\chi^2$ test (e.g., Galbraith, 2005), the uncertainty of the pooled age may be expanded to reflect the actual inter-grain age variation (see Eq. (C6) in Appendix C) or the data may be treated as scattered (sect. 3.4.6). If there exists a relationship between age and grain properties, e.g. by plotting the age against these grain properties or to color-coding the D-P plot (e.g., Fig. 7b), the age distribution may be described by a central age or, if possible, a thermal-history model accounting for the effect of the specific grain property on age.

### 3.4.3 Linear relationship with systematic offset

Systematically offset data must be treated with caution as such data pose problems for many common data-analysis tools (see sect. 2.3). The only sample age that may appropriately describe systematically offset data is the isochron age determined from the slope of a regression line (see Fig. 4b; section 2.3; Appendix C2). Another option is to verify the reason of the intercept, such as zoning, 'parentless helium', or a counting bias and to develop an analytical strategy to eliminate the bias (e.g., Spiegel et al., 2009; Orme et al., 2015). The intercept of the regression line provides a first-order estimate for the amount of offset. If the intercept is large, close to the mean daughter concentration, or if the data allow for a horizontal or vertical line fit, they could also be treated as a cluster (sect. 3.4.2) or as scattered data (sect. 3.4.6). If the data are over-dispersed, e.g., showing an MSWD outside its confidence interval, it is possible to expand the uncertainty on the isochron age by multiplying it with $\sqrt{\left( MSWD \right)}$ (e.g., Ludwig, 2012). For a strong overdispersion (e.g., MSWD>10), the data should be treated as scattered (see sect. 3.4.6).

### 3.4.4 Non-linear relationship

A non-linear relationship in the D-P plot points to radiation-damage-dependent daughter retention. This assumption can be tested against independent radiation-damage measurements. Raman and infrared spectroscopy, or X-ray diffraction provide radiation-damage estimates for zircon or titanite (e.g., Nasdala et al., 1995; Deliens et al., 1977; Holland and Gottfried, 1955; Heller et al., 2019), while optical absorption or Raman spectroscopy are potential tools to measure radiation damage in apatite (e.g., Ritter and Märk, 1984; Liu et al., 2008). Alternatively, a non-linear D-P relationship could result from daughter retention depending on other grain properties and the different grains recording the same thermal history differently. This effect can be examined by plotting the age against these parameters or by color-coding the D-P plot (e.g., Fig. 7b). If such a relationship exists, the dataset may be described by a central age (see Appendix C). If the decision for a non-linear versus a linear relationship with an offset is not clear (E in Fig. 11; Fig. 12), the less complex linear model should be preferred over a non-linear model (sect. 3.4.3) in the absence of independent radiation-damage measurements. For a non-linear trend caused by radiation-damage-dependent daughter retention, forward modeling of daughter retention and radiation-damage accumulation and annealing provides further insights into a sample's thermal history (e.g., Flowers et al., 2009; Willett et al., 2017; Guenthner et al., 2013). In this case, the D-P plot allows to compare the data to the D-P relationship predicted by the model, especially in the low-eU region, where the model prediction connects to the origin (Härtel et al., 2022a). Figures 5b, 7c, and 8b show thermal-history forward models for zircon He dating plotted as lines in the D-P plot.

### 3.4.5 Several populations

If the D-P plot suggests that several discrete age components are present in the sample, the KDE or radial plot are the standard tools to examine the data. The occurrence of different components should also be tested for consistency, e.g., if a

mixture of populations makes sense in the geological context (sect. 3.1) or by color-coding according to a variable that may underlie the different populations (see Fig. 6b). The age distribution can either be described by a finite-mixture model (e.g., Galbraith and Green, 1990; Galbraith and Laslett, 1993; Galbraith, 2005; Vermeesch, 2019) or by separating the data into age populations to be analyzed individually according to the procedure in Fig. 11.

### 3.4.6 Scattered data

Data that vary strongly in age and are scattered in the D-P plot may result from several scenarios: First, they may be a consequence of underestimating the uncertainties with respect to the variation in the single-grain data (e.g., for He dating, Fitzgerald et al., 2006; Brown et al., 2013). Martin et al. (2023) and Zeigler et al. (2023) showed that especially the uncertainty related to α-ejection correction in whole-grain He dating is difficult to estimate, while the correction contributes significantly to the age error. Data with limited scatter, for which the uncertainties may be underestimated may be treated as a cluster (sect. 3.4.2). A second explanation for scatter is the occurrence of different age populations, which can be verified in a KDE plot (sect. 3.4.5). Third, the scatter may also be due to each grain having slightly different daughter-retention properties and recording a different age. Plotting the age against these parameters or color-coding the D-P plot (Fig. 7b) allows to assess this relationship; a central age may be used to describe such a continuous mixture (sect. 3.4.2; Appendix C). If the scatter cannot be explained by one of these scenarios (e.g., Fig. 7c), the range of the single-grain ages should be reported. Scattered data also pose a serious problem to inverse time-temperature (t-T) modeling, as the age difference may not allow for a single t-T path to reconcile the spread in ages.

### 3.4.7 Inverse relationship

An inverse daughter-parent relationship runs contrary to the relationship expected from the age equation (Fig. 8). In general, two scenarios can account for this relationship without pointing to an analytical problem. If the dataset is small (e.g., n≤5), a spurious inverse trend could arise randomly (Ketcham et al., 2018) and the dataset should be treated as scattered (sect. 3.4.6). However, the interpretation of small datasets should be carried out with caution (see sect. 4). Alternatively, the inverse relationship may represent an inverse segment of a non-linear trend if radiation damage controls daughter retention (sect. 3.4.4; Fig. 8b). If there is no clear explanation for the inverse daughter-parent relationship, it is best to report the range of single-grain ages (Appendix C).

### 3.5 Practical limits of D-P plotting

The data-analysis workflow in Figure 11 provides simple decision paths and criteria for assigning a dataset to a class. This has the advantage of keeping the data-analysis process consistent, especially for studies involving many samples. Still, this decision-based approach has some limits that we would like to point out.

First, our ability to evaluate the D-P relationship for a sample clearly depends on the number of data. There are several limits a small sample imposes on data analysis using the workflow in Figure 11: (1) it is not possible to recognize different populations; (2) a single outlier may constitute a large proportion of the gathered data; (3) random variation may cause inverse D-P relationships (see Ketcham et al., 2018) or spurious associations between the age and other properties; (4) in terms of sample ages, the small number of grains inhibits the use of isochron or central ages, which would require the fitting of several parameters (age and intercept or dispersion) to a small amount of data. While this hampers a strict classification following Fig. 11, it is still possible to use the D-P plot as a qualitative guide, e.g., to visualize the data in terms of their variation in D and P. It also enables to examine, in which D-P direction a potential outlier deviates from the rest of the data. For example, this helps to decide if the pooled age is biased towards a single high-D or -P grain (see Appendix C1). In this case, we recommend to check this potential outlier or report the single-grain age range. The number of analyzed grains is not a concern for FT and ZR dating (n>10), but it is a limiting factor for conventional whole-grain He dating (n<10). However,

the recent development of laser ablation based He dating will increase the number of grains analyzed per sample and

recognizing D-P relationships (e.g., Tripathy-Lang et al., 2013; Pickering et al., 2020). In addition, some cases may allow grouping together data from several small samples. This approach hinges on the condition that the different samples are comparable, e.g., that they share the same thermal history in the partial annealing/retention zone of the used thermochronometer. This strategy is often used for analyzing He data with respect to radiation-damage effects (e.g., Figs. 7b, 8b; Guenthner et al., 2017; Baughman et al., 2017; Ault et al., 2018; Armstrong et al., 2024).

Second not all datasets may be assignable unambiguously to a class. Examples may be cases of moderate variation falling between clustered or scattered data or cases, in which the distinction between linear- and non-linear relationships is not clear (see Fig. 12). While section 3.4 provides suggestions for alternative classifications, this problem highlights the necessity for transparent reporting on the decisions taken by the analyst (Tab. 1).

Second, unreset or partially reset detrital samples often record a complex mixture of pre- and post-depositional thermal
history. They also often contain grains with different chemical composition and size. Detrital samples are therefore not expected to fit into the simple categories of Fig. 11. Extracting a sample age or an interpretation from a single sample or a single thermochronometer is usually not possible (e.g. Carter, 2019). Standard procedures for interpreting detrital thermochronological data include identifying peak ages in the single-grain age distribution and putting them into the context of the stratigraphic age, age distributions of source areas, catchment geometry, etc. (e.g., Malusà and Fitzgerald, 2019).
While it is possible to evaluate different age populations in the D-P plot (see sect. 3.4.5), KDE or radial plots are the more adequate tools for this task. Still, the D-P plot may hold additional information that is difficult to access with these plots. First, it may be used on a subset of the data to evaluate the daughter-parent relationship for a given age population and possibly detect a non-linear or systematically offset relationship (sect. 3.4.3, 3.4.4). However, this can only be done reliably, if enough data (e.g., n ≥10) are available in this grain population. Second, it may help to identify bias in grain selection. One
of these is the problem with overlapping, uncountable fission tracks in old or U-rich zircon, that may skew ZFT age populations towards younger ages and thus affect the interpretation in terms of source-area exhumation and erosion patterns (e.g., Malusà, 2019).

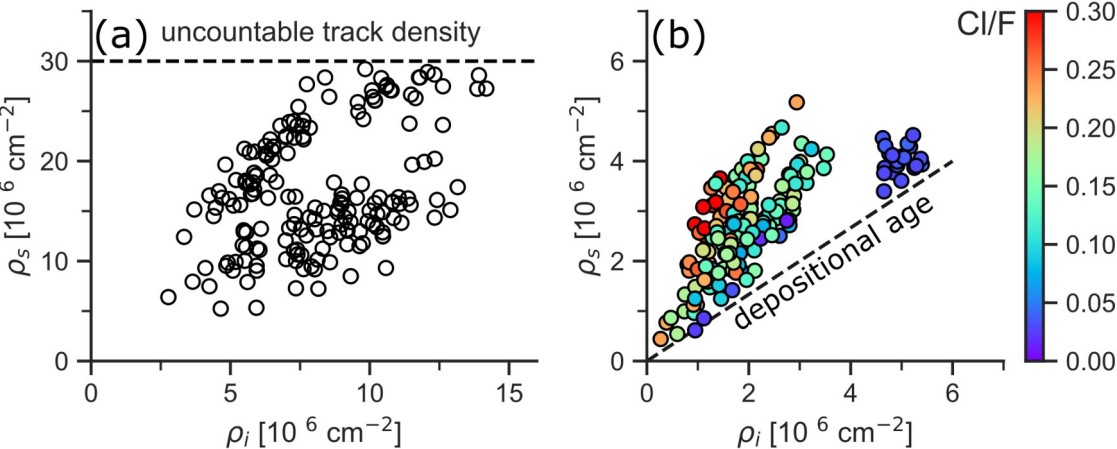

**Figure 13. Possible applications of D-P plots for detrital thermochronology. (a) D-P plot for a synthetic ZFT dataset with the dashed line marking the density threshold, at which the spontaneous tracks become uncountable. (b) D-P plot for a synthetic AFT dataset color-coded by the Cl/F ratio. The dashed line represents the depositional age.**

Figure 13a shows the D-P plot for a synthetic ZFT dataset. The dashed line marks the countability limit for the spontaneous tracks. This limit cuts off the track-density distribution for an old grain population; it indicates that the sample may contain
older or higher-U grains not datable with the ZFT method. A third application is the visualization of different grain populations with respect to age, parent concentration and other grain properties, e.g. grain size or composition to highlight

nuances in the composition of different age populations. Figure 13b shows the D-P plot for a synthetic AFT dataset, color-coded by the Cl/F ratio and with a dashed line representing the depositional age. In this case, part of the grains in the age group slightly older than the depositional age stands out due to its high induced-track density (high U content) and its F-dominated halogen composition. So, despite the complexity of detrital samples, there are situations, in which the visualization of the data in a D-P plot can be useful.

## 3.6 D-P plotting in Incaplot

This section briefly describes Incaplot (Härtel, 2024), a simple, Python-based graphical-user-interface software dedicated to producing D-P plots. Existing softwares (e.g., Trackkey, Isoplot Excel, IsoplotR) already provide the tools for D-P plotting, but these are often buried between other functions or are available for certain dating methods only. Incaplot is available for free at https://zenodo.org/records/8233941 as a one-file executable for Mac (MacOS 10.15 Catalina and younger) and Windows operation systems (Windows 8 and younger).

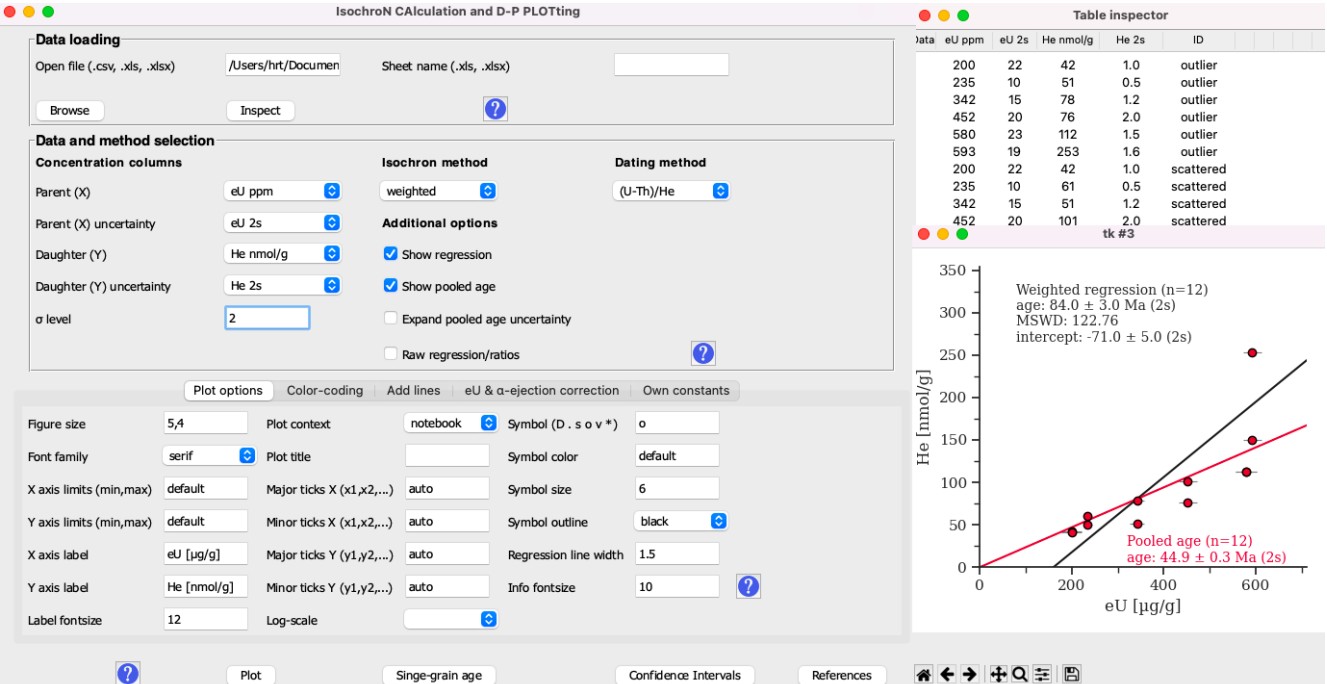

**Figure 14. Main window of the Incaplot software (left), the table inspection tool (upper right) and an output D-P plot (lower right).**

Incaplot allows to create D-P plots and calculating low-temperature themochronometric ages. It also provides a range of visualization and customization options. Figure 14 shows Incaplot's main window (left), its data inspection tool (upper right) and its graphical output (lower right). The main window consists of three frames dedicated to (1) loading data files, (2) the input data and calculation algorithms to be used, and (3) modifying the plots and calculations.

Incaplot requires the input files to be Excel spreadsheet files in .xls, or .xlsx format or comma-separated (.csv) text files. The plotting variables need to be organized as columns with the variable names in the first row. A user manual for the current Incaplot version and an example file displaying the input data format are available in Incaplot's zenodo repository.

Incaplot provides a range of plot-customization options, which include customizing markers, axes and ticks, adding line segments to plots, and color-coding plots by discrete and continuous variables. While Incaplot was set up to handle mainly He, ZR and FT data, it can also be used for other dating systems or generic scatterplots. The output plots are exportable in different raster (.jpg, .png, .tif) and vector formats (.svg, .pdf, .eps).

Besides D-P plotting, Incaplot contains functions for sample-age calculation as pooled age, isochron fitting with different algorithms (see Appendix C2), calculation of single-grain ages and effective uranium concentrations (see equation (1) and Appendix A).

4 Conclusions

A plot of daughter *vs.* parent concentration (D-P plot) represents a graphical solution of the age equation in radiometric dating and is an effective tool to reveal information in low-temperature thermochronology data. Its unique advantages are its capabilities to detect systematic offsets or radiation-damage effects in the data, which often compromise other data-analysis tools. It also enables the analyst to identify potential outliers with respect to both, daughter and parent concentration rather than the single-grain age only. These advantages make it an ideal first step for data analysis, allowing us to adapt the analysis strategy to a given data pattern. We show several published datasets exemplifying the range of possible D-P relationships and the underlying geological factors, and propose a new workflow for using D-P plots in thermochronological data analysis. Our approach follows a step-wise examination of the daughter-parent relationship and assigns one of seven classes to it. Based on the daughter-parent relationships, it provides criteria to choose further data-analysis tools and – if appropriate – calculate a sample age. The classification scheme is an attempt to make data analysis more consistent and transparent. Our classification approach has limitations, especially when applied to small or detrital datasets, however, the D-P plot itself may still provide relevant insights in these cases. We also introduce Incaplot, a free, graphical-user-interface software and invite everyone for creating and customizing D-P plots in a straightforward way.

## Appendix A: The effective uranium concentration

The effective uranium concentration (eU) is a summary of the α-producing U, Th, and Sm concentrations, rescaling them to a common decay rate of U:

$$eU = k_U [U] + k_{Th} [Th] + k_{Sm} [Sm] \tag{A1},$$

with the terms in the brackets being the concentrations in units of mass, and $k_U$, $k_{Th}$ and $k_{Sm}$ being coefficients for each concentration. There are currently two definitions of eU that result in slightly different coefficients. Shuster et al. (2006) and Cooperdock et al. (2019) recalculate the actinide concentrations to a concentration of total U, whereas Härtel et al. (2021) recalculate them to the decay rate of $^{238}$U only. The latter approach enables us to use eU as a single parent with a well-defined decay rate for He and ZR dating. It also considers the change of the daughter-production rate over geological time instead of using present-time production rates. Härtel et al. (2023) showed that the formulation

$$eU = 1.05 [U] + 0.24 [Th] + 0.0012 [Sm] \tag{A2}$$

gives accurate results for samples at 30<t<1000 Ma, but may be modified if the expected ages for a set of samples are consistently higher or constrained well-enough to calculate them more accurately.

The coefficients for eU are derived in Eq. (A3)–(A9). The starting point is the α-production equation:

$$N(\alpha) = 8 \frac{N_A \left[ {}^{238}U \right]}{M_{238}} \left( e^{\lambda_{238} t} - 1 \right) + 7 \frac{N_A \left[ {}^{235}U \right]}{M_{235}} \left( e^{\lambda_{235} t} - 1 \right) + 6 \frac{N_A \left[ {}^{232}Th \right]}{M_{232}} \left( e^{\lambda_{232} t} - 1 \right) + \frac{N_A \left[ {}^{147}Sm \right]}{M_{147}} \left( e^{\lambda_{147} t} - 1 \right) \tag{A3}.$$

$N(\alpha)$ is the number of alpha decays, $N_A$ is the Avogadro constant, 8, 7, 6 and 1 are the numbers of alpha particle produced by the respective decay series, M are the molar masses, and $\lambda$ the decay constants and the symbols in brackets the concentrations in units of mass. The constants used in the calculations are summarized in Table A1. Rescaling all summands to the terms of $^{238}$U gives:

$$N(\alpha) = 8 \frac{N_A}{M_{238}} \left( e^{\lambda_{238} t} - 1 \right) \left[ \left( 1 + \frac{7 M_{238} \left( e^{\lambda_{235} t} - 1 \right)}{8 M_{235} \left( e^{\lambda_{238} t} - 1 \right)} \frac{w_{235}}{w_{238}} \right) \left[ {}^{238}U \right] + \left( \frac{6 M_{238} \left( e^{\lambda_{232} t} - 1 \right)}{8 M_{232} \left( e^{\lambda_{238} t} - 1 \right)} \right) \left[ {}^{232}Th \right] + \left( \frac{M_{238} \left( e^{\lambda_{147} t} - 1 \right)}{8 M_{147} \left( e^{\lambda_{238} t} - 1 \right)} \right) \left[ {}^{147}Sm \right] \right] \tag{A4}.$$

This equation can be simplified by replacing the weighted actinide concentrations in the square brackets by eU:

$$N(\alpha) = 8 \frac{N_A}{M_{238}} \left( e^{\lambda_{238} t} - 1 \right) [eU] \tag{A5}.$$

This results in:

$$eU = \left[ \left( 1 + \frac{7 M_{238} \left( e^{\lambda_{235} t} - 1 \right)}{8 M_{235} \left( e^{\lambda_{238} t} - 1 \right)} \frac{w_{235}}{w_{238}} \right) [U] w_{238} + \left( \frac{6 M_{238} \left( e^{\lambda_{232} t} - 1 \right)}{8 M_{232} \left( e^{\lambda_{238} t} - 1 \right)} \right) [Th] + \left( \frac{M_{238} \left( e^{\lambda_{147} t} - 1 \right)}{8 M_{147} \left( e^{\lambda_{238} t} - 1 \right)} \right) [Sm] w_{147} \right] \tag{A6}.$$

$w_{235}$, $w_{238}$ and $w_{147}$ are the mass fractions of the $^{235}$U, $^{238}$U and $^{147}$Sm isotopes and the terms in square brackets are element concentrations. Equations (A7)–(A9) define the coefficients in (A1) for each element:

$$k_U = w_{238} + \frac{7 M_{238} \left( e^{\lambda_{235} t} - 1 \right)}{8 M_{235} \left( e^{\lambda_{238} t} - 1 \right)} w_{235} \tag{A7},$$

$$k_{Th} = \frac{6 M_{238} \left( e^{\lambda_{232} t} - 1 \right)}{8 M_{232} \left( e^{\lambda_{238} t} - 1 \right)} \tag{A8},$$

$$k_{Sm} = \frac{M_{238} \left( e^{\lambda_{147} t} - 1 \right)}{8 M_{147} \left( e^{\lambda_{238} t} - 1 \right)} w_{147} \tag{A9}.$$

Figure A1 shows how the normalization coefficients for each α-producing element change with respect to the age of a sample.

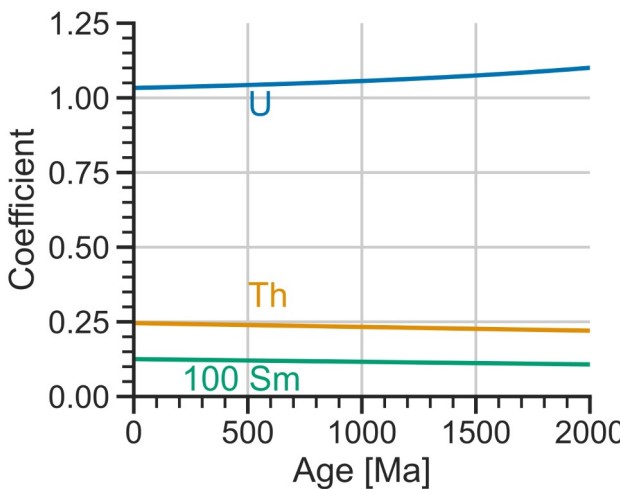

Fig. A1. Time-dependence of the coefficients for U, Th and Sm
(multiplied by 100) in the eU equation (2).

The time-dependence in Eqs. (A7)–(A9) also allows iterative age calculation for He and ZR dating. This requires calculating eU from equation (A2) and then alternating between calculating the age from Eq. (1), and recalculating eU from Eqs. (A1) and (A7)–(A9) until the solutions converge.

Table A1. Coefficients and constants used in the calculations. The atomic masses and mass abundances are based on Holden et al. (2018), the decay constants are from Jaffey et al. (1971), Steiger and Jäger (1977), and Holden (1990). The decay constants are rounded to the first significant digit of their uncertainty.

| Constant | Value |
| --- | --- |
| $\lambda_{238}$ | $1.551\ 10^{-10}$ a$^{-1}$ |
| $\lambda_{235}$ | $9.848\ 10^{-10}$ a$^{-1}$ |
| $\lambda_{232}$ | $4.95\ 10^{-11}$ a$^{-1}$ |
| $\lambda_{147}$ | $6.5\ 10^{-12}$ a$^{-1}$ |
| $M_{238}$ | 238.05 g/mol |
| $M_{235}$ | 235.04 g/mol |
| $M_{232}$ | 232.04 g/mol |
| $M_{147}$ | 146.91 g/mol |
| $N_A$ | $6.022\ 10^{23}$ mol$^{-1}$ |
| $w_{235}$ | 0.0072 |
| $w_{238}$ | 0.9928 |
| $w_{147}$ | 0.1466 |

## Appendix B: Units of daughter and parent concentrations

Daughter and parent concentrations can be expressed differently in external-detector-method FT and whole-grain He dating. Several criteria can be considered to find the right set of units for the D-P plot.

In He dating, the pairs of daughters (He) and parents (eU from U, Th, Sm) can either be expressed in units of abundance and mass (e.g., fmol and ng) or as concentrations (e.g., nmol/g and µg/g). The difference between these units is the normalization by the mass of the analyzed grain. For non-normalized data, the size or mass of the analyzed grains will introduce variation

into D and P that is unrelated to the age of the sample. In case the grains differ strongly in size, this may bias the pooled age towards the largest grains and the isochron age towards the smallest or the largest ones (see Appendix C). Rescaling the units of D and P to concentrations eliminates this potential bias. Furthermore, it is advantageous to correct the He concentration for α-ejection correction before calculating the age: correcting for α-ejection after age calculation introduces a positive bias to the age (e.g., Vermeesch, 2008). Therefore, the corrected He concentration should be used as daughter concentration for

plotting. In external-detector FT dating, a similar question of units arises concerning the use of either the spontaneous- and induced-track counts or their track densities. In this case, it is advantageous to use the track densities instead of the counts to avoid bias towards big grains.

The specific units then determine the value of the constant c in Eq. (1). Re-arranging it to a daughter-production equation gives:

$$D = \frac{1}{c}\left(e^{\lambda t} - 1\right)\left[P\right] \tag{B1}.$$

For ZR dating, c results from equating Eq. (B1) and (A8):

$$c = \frac{M_{238}}{8\,N_A} = 4.94\,10^{-23}\,g/\alpha \tag{B2}.$$

Given input damage densities in $10^{16}$ α/g and eU concentrations in µg/g, c takes a value of 0.494 [$10^{-16}$ µg/α].

For He dating, the same relationship as for ZR dating applies, with the difference of He concentrations usually being

reported in molar concentrations:

$$c = \frac{M_{238}}{8} = 29.76\,g/mol \tag{B3}.$$

If the input He concentrations are in in nmol/g and the eU concentrations in µg/g, c takes a value of 0.02976 [µg/nmol].

For FT dating, the constant c depends on measured experimental factors. This gives:

$$c = 0.5\,\lambda_D\,\zeta\,\rho_D \tag{B4}$$

for the external detector method, where 0.5 is the geometry factor, $\lambda_D$ is the total decay constant for $^{238}$U, $\zeta$ is the proportionality factor determined from dating an age reference material, and $\rho_D$ the dosimeter track density (see Hurford, 2019). In this case, c is dimensionless because the spontaneous and induced-track counts densities are expressed in the same measurement units.

Laser-ablation FT dating requires a slightly different value for c because no dosimeter glass is involved in parent

measurement (see Vermeesch, 2019):

$$c = 0.5\,\lambda_D\,\zeta \tag{B5}.$$

In this case, the dimension of c depends on the units of parent measurement, e.g. as U concentration or as element ratio, e.g., U/Ca.

## Appendix C: Age calculation and reporting

### C1 Mean ages

For datasets showing a single age, it is attractive to report the arithmetic mean age due to its familiarity and simple calculation. However, the mean age is inadequate for summarizing most thermochronological ages. First, calculating a mean from ages determined by a logarithmic age equation as in (1) 'linearizes' the age equation and causes a negative bias compared to applying the logarithmic age equation to a mean D/P ratio. Second, even when directly applied to the ratio, the arithmetic mean gives a biased age estimate, as can be shown from its relationship to the pooled age (see below; Pearson, 1896; Härtel et al., 2022a):

$$t_{mean} = t_{pooled}\left(1 - r_{DP}\, v_P\, v_D + v_P{}^2\right) \tag{C1}.$$

$v_D$ and $v_P$ are the variation coefficients (standard deviation divided by arithmetic mean) of the daughter and parent concentrations, and $r_{DP}$ is their correlation coefficient. Equation (C1) shows that for the ideal proportional D-P relationship ($r_{DP} = 1$, $v_D = v_P$), the mean and pooled ages are the same. In a less ideal case, the measurement error on the parent concentration increases $v_P$ and – as it is independent of the daughter concentration – weaken the relationship between D and P (decreasing $r_{DP}$). This causes the mean age to increase with respect to the pooled age. It means that the mean age is biased towards higher ages under non-ideal daughter-parent relationships. This is especially problematic for the whole-grain He and laser-ablation FT methods, for which the analytical uncertainties are often too small to explain the observed age variation (e.g., Fitzgerald et al., 2006; Ketcham et al., 2018). Essentially, measurement error on the parent concentration creates a right-skewed age distribution, whose mean increases with increasing variance and is biased towards higher ages.

A more robust alternative for calculating a central tendency is the pooled age, i.e., treating all analyzed grains as a single grain by summing up all daughter and parent concentrations. The age is then calculated by substituting the ratio of these sums for D/P in Eq. (1):

$$t_{pooled} = \frac{1}{\lambda}\ln\left(1 + c\,\frac{\sum D}{\sum P}\right) \tag{C2}.$$

Vermeesch (2008) pointed out that in the presence of outliers with high parent concentration or age, the pooled age is biased towards these grains. Also, Green (1981) and Galbraith and Laslett (1993) argued that the pooled age is not appropriate as sample age, if the age variation cannot be explained by the estimated uncertainties. However, in the case of clustered data (sect. 3.4.2) or those forming a linear trend with zero intercept (sect. 3.4.1) without outliers, the age variation is small so that the bias on the pooled age can be assumed to be negligible. The uncertainty on the pooled age can be estimated from error propagation of the single-grain uncertainties. For He and ZR dating, this gives:

$$s\left(t_{pooled}\right) = t_{pooled}\sqrt{\frac{\sum s(D)^2}{\left(\sum D\right)^2} + \frac{\sum s(P)^2}{\left(\sum P\right)^2}} \tag{C3},$$

with s representing the uncertainties on D, P, and t, respectively. FT dating requires to also take into account the uncertainty on c in Eq. (C2). For the EDM method, this gives (Galbraith, 2005):

$$s\left(t_{pooled}\right) = t_{pooled}\sqrt{\left(\frac{s(\zeta)}{(\zeta)}\right)^2 + \frac{1}{\sum N_s} + \frac{1}{\sum N_i} + \frac{1}{\sum N_d}} \tag{C4}.$$

$N_s$, $N_i$ and $N_d$ are the spontaneous, induced, and dosimeter track counts, respectively; $\zeta$ and $s(\zeta)$ are the calibration factor and its uncertainty.

For laser-ablation FT dating, the uncertainty on the pooled age is:

$$s\left(t_{pooled}\right) = t_{pooled}\sqrt{\left(\frac{s(\zeta)}{(\zeta)}\right)^2 + \frac{1}{\sum N_s} + \frac{\sum s(P)^2}{\left(\sum P\right)^2}} \tag{C5}.$$

If the ages from a dataset are over-dispersed due to the uncertainties not reflecting the variation in the data, it may be advantageous to estimate the uncertainty of the pooled age directly from the variation in D and P concentrations (e.g., Pearson, 1896):

$$s(t) = t \sqrt{\frac{v_D^2 + v_P^2 - 2 r_{DP} v_D v_P}{n}} \tag{C6}.$$

$v_D$ and $v_P$ represent the variation coefficients of D and P, and $r_{DP}$ is the correlation coefficient for the D-P relationship. Equation (C6) may give a more realistic uncertainty estimate than those in Eq. (C3)-(C5) if the data are slightly over-dispersed. For strongly scattered data, however, (C6) gives a large uncertainty, confirming that a single sample age may be meaningless.

## C2 Isochron ages

For systematically offset data (sect. 3.4.3), the single-grain ages and the pooled age are offset in the same direction and give erroneously high or low ages (see sect. 2.3). In this case, it is advantageous to calculate an isochron age by fitting a regression line to the D-P data and replacing D/P in Eq. (1) by the slope m:

$$t_{isochron} = \frac{1}{\lambda} \ln(1 + cm) \tag{C7}.$$

The uncertainty on the isochron age results from propagation of the slope's uncertainty. This logarithmic age equation avoids the bias of the isochron age identified by Vermeesch (2008) for a linear age equation.

Typical algorithms for fitting isochrons are uncertainty-weighted (York, 1968; Kullerud, 1991) and robust regression (Huber, 1981; Powell et al., 2020). Both of these assign weights to each data point: the former based on the measured uncertainty, the latter based on the uncertainty and the distance of each point from a linear 'spine' in the data. Robust regression is therefore useful for datasets in which single grains fall off well-defined trends. However, its benefits are limited in the case of many grains deviating from the trend. These regression algorithms, together with the classic least-squares regression are implemented in Incaplot.

In general, data at the low- and high-parent ends of the distribution and data with small uncertainties have a strong influence on the isochron age, making it sensitive for outliers. Its use should therefore be limited to cases of systematic offset in the D-P relationship. Apart from the isochron age, the intercept may also contain important information for the interpretation and should be reported together with the age (sect. 3.4.3).

The mean square weighted deviation (MSWD; or the spine width for robust isochrons) of the isochron provides information on how well the isochron fits the data. An MSWD within the confidence interval (Table C1) indicates that the variation of the data about the isochron is within the range expected from the input uncertainties. A high MSWD outside the confidence interval (Table C1) denotes over-dispersed data, whose variation is not explained by the input uncertainties alone – this may either point to unidentified sources of error or inter-grain variation of true ages within a sample. For He and laser-ablation FT data, whose sources of error are not yet well understood, these metrics have to be used with caution.

A standard practice to account for over-dispersed data in geochronology is to expand the uncertainty of the isochron age, multiplying it by $\sqrt{MSWD}$ (e.g., Ludwig, 2012).

**Table C1. Confidence intervals (95 %) for the MSWD and the spine width for isochron fits (n-2 degrees of freedom). The MSWD intervals are based on Wendt and Carl (1991), the intervals for the spine width are from Powell et al. (2020).**

| | MSWD | | Spine width | |
| --- | --- | --- | --- | --- |
| n | Lower boundary | Upper boundary | Lower boundary | Upper boundary |
| 10 | 0.50 | 2.00 | 0.31 | 1.55 |
| 15 | 0.61 | 1.78 | 0.4 | 1.5 |
| 30 | 0.73 | 1.53 | 0.58 | 1.39 |
| 60 | 0.81 | 1.37 | 0.71 | 1.28 |

**C3 Age mixtures**

Apart from the simple cases, discrete or continuous mixtures of ages may occur. There are two strategies to deal with discrete age components in a sample (sect. 3.4.5): mixture modeling (e.g., Galbraith and Laslett, 1993; Galbraith, 2005; Vermeesch, 2019), or splitting the data into different groups and calculating sample ages for each of them.

A continuous age mixture occurs if a sample contains grains with a wide range of kinetic properties responding differently to same thermal history (e.g., Vermeesch, 2019) – each grain then acts as single thermochronometer. An example could be the apatite FT age in a monotonously cooled plutonic rock with grains of different Cl/F ratio. In this case, the intra-sample age variation reflects both, the measurement error and the true-age variation between grains. This distribution is best described by a 'random effects model' and the age to be reported is the central age (Galbraith and Laslett, 1993) – the dispersion parameter describes the variation in true ages. Note however, that it is necessary to relate the single-grain age to a kinetic parameter such as grain size, mineral chemistry, or measured radiation damage (Fig. 7b) to justify the use of a continuous mixture of ages. Galbraith (2005) and Vermeesch (2019) provide further discussion and calculation algorithms of the central age for FT dating, and Vermeesch (2008) for He dating. For complex data that cannot be described by a discrete or continuous mixture, we suggest to report the range of single-grain ages, which requires no additional assumptions.

**Author contribution**

BH: Conceptualization, Methodology, Formal analysis, Visualization Writing – original draft preparation; EE: Conceptualization, Visualization, Funding acquisition, Writing – review and editing.

**Competing interests**

The authors declare that they have no conflict of interest.

**Acknowledgements and Funding**

This research was funded by the University of Calgary Eyes High postdoctoral match funding to Birk Härtel, the Natural Sciences and Engineering Research Council of Canada (NSERC) RGPIN-2024-03863 (Eva Enkelmann) and the American Chemical Society – PRF # 67107-ND8 (Eva Enkelmann). We thank Stephen Cox and an anonymous referee for their thoughtful reviews that greatly improved our manuscript. We also thank Tibor Dunai for editorial handling.

**Code/Data availability**

The synthetic D-P data shown in Figs. 1-8 are available as a supplementary file to this article. Incaplot is available as standalone executable for MacOS and Windows OS and as Python code at https://zenodo.org/records/8233941.

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
