# Peer review of "The daughter-parent plot: a tool for analyzing thermochronological data"

_Geochronology, 2024_

## Author Response (AR1)

**Manuscript gchron-2024-1 - Revision 1**

**The daughter-parent plot: a tool for analyzing thermochronological data**

Birk Härtel and Eva Enkelmann

————————————————————————————————————————————

Dear Editor,

we appreciate the review comments by Dr. Stephen Cox and an anonymous reviewer, as they helped us redefining the scope of our manuscript and making some significant improvements to the text. We are pleased that neither of the reviewers found a technical flaw in the manuscript. Following the suggestions of the reviewers, we made the following major changes to our manuscript: (1) we clarified the scope of our manuscript as providing guidelines for using the daughter-parent (D-P) plot, which is currently not used to its full potential in low-temperature thermochronology; (2) we emphasize the difference between the D-P and the isochron plot; (3) we include several examples for daughter-parent relationships from published datasets to discuss the origin of specific data patterns; (4) we point out the unique advantages of the D-P plot over other data-analysis tools; and (5) we introduce Incaplot, a free software that facilitates creating D-P plots in order to make the tools presented in our manuscript more accessible. In our opinion, these modifications have strongly improved our manuscript, and we feel that the revised manuscript can be reconsidered for publication. Please find below the specific replies to the reviewer comments.

Kind regards,

Birk Härtel for the co-authors

**Referee #1 (Stephen Cox)**

We appreciate the review comments by Stephen Cox, whose suggestions will significantly improve our manuscript. We are pleased that he does not contest the theoretical concept of the workflow, nor does he point out any technical mistake. However, there seems to be a misunderstanding regarding the scope and purpose of this contribution, and he suggests to include examples of real-world datasets to improve the impact and show the applicability of the D-P plot. Below we respond to each of these major and all minor comments in detail. **R1** marks the reviewer#1 comments and **A** the authors' replies.

R1: This submission is designed to argue that isochron plots, rather than age-eU plots, should be used to interpret compositional trends in thermochronological data.
This point has been made before, notably by Vermeesch (2008) and by this same lead author in a paper published in December 2022. Aside from some simulated data plots that might have been beneficial to bolster the theoretical arguments made in the 2022 paper, it is not clear to me what innovation is presented in the present work.

A: We see a misunderstanding here, regarding the scope and motivation of our manuscript. It is not our aim to replace the age-eU plot by the D-P plot or to repeat the work of Vermeesch (2008) or Härtel et al. (2022). While these publications are concerned with applying the D-P plot to specific problems, we look at the bigger picture of making sense of thermochronological data in general. The mentioned publications are theoretical and mathematics-heavy and for that reason overlooked by the many users of thermochronometric data that are mostly interested in answering their geological question. We think that the D-P plot sits at the important interface between the analytical results and more specific data-analysis tools, such as radial plots or age vs. grainsize plots. However, there is a gap in thermochronological literature on how to use the D-P plot for data analysis towards a geological interpretation.
Our aim is to fill this gap and provide guidance to an audience that is less knowledgeable on the analytics of laboratory data such as Dr. Cox. Our own experience from providing low-temperature analytical services for others (university collaborators and students, geological surveys, and industry) has shown that many of them feel lost and confused how to proceed when given their data. The simply referral to publications such as Vermeesch (2008), Härtel et al. (2022) is not sufficient. Thus – the purpose of this manuscript is to target this "user" audience, which is significantly large, and describe how to use patterns in the D-P plot for data analysis and linking its use to sample-age calculation and the use of more common data-analysis tools (e.g., Flowers et al., 2022a, b ; Kohn et al., 2024).

This comment made us aware that this goal and target audience has not been stated and we revised our introduction section to clearly state our motivation. (see also responses and revisions below that are targeted towards this goal)

R1: I would request that the authors clearly explain the following in detail: Why are we calling these "daughter-parent plots" and not isochron plots, as they have been known in the literature for decades? Like the similar "isotope correlation diagram," this seems unnecessary and far more likely to cause confusion than the technical clarification that the authors presumably intend.
The burden placed on the person who would request a change scales with the preponderance of contrary historical and common usage. In this case, the burden is a heavy one and the authors do not even engage with it.
At the very least, it should be openly acknowledged that this is just a name change for an extremely familiar tool, otherwise it seems like this is designed to confuse the reader into thinking this is something new.

A: We understand that there is a strong similarity between the D-P and the isochron plot, which might be confusing to readers familiar with the concept from radioisotopic dating, which is the background of Dr. Cox.
We prefer to stick to the name 'daughter-parent plot' due to the following reasons: (1) technically, the D-P plot is not the same as classic isochron plot. In contrast to the D-P plot, the latter assumes the initial presence of the radiogenic daughter isotope by default, and its x and y data are same-denominator ratios with error correlation. (2) In terms of the intended audience, our paper mainly aims at the low-temperature thermochronology community, in which such a plot is not a common tool, and when used, there is no consensus on calling it "isochron plot" (Fanale and Kulp, 1962; Green, 1981; Wernicke and Lippolt, 1993; Galbraith, 1997; Dunkl, 2002; Vermeesch, 2008; Hueck et al., 2018; He et al., 2021; Meier et al., 2024). (3) from a practical perspective, the name "isochron plot" emphasizes fitting a line to the data and reporting an isochron age. This is not a good use of the plot in cases where the data do not define a linear trend but where the D-P plot is still a useful tool for characterizing the age variation. (4) the name daughter-parent plot is very descriptive of what is plotted and thus easily accessible for users.
We agree with the reviewer that this naming may cause confusion for those with a background in geochronology and using isochron plots, for that reason, we will revise our manuscript and explain the apparent similarities and differences between the D-P and the isochron plot.

R1: How does this work expand meaningfully on the arguments made in Härtel et al., 2022 and already largely covered by previous work (e.g., Vermeesch 2008)? Generating some new plots from simulated data does not constitute an advance that merits an additional paper. Using a longer format journal as an extended discussion section or appendix for another publication is bad practice.

A: This manuscript focuses on the more general problem of analyzing thermochronological data from D-P relationships. We are not aware of any publication detailing an interpretation strategy for thermochronological data using the D-P plot as our Figure 2 and section 3.3, both of which are uncontested by Dr. Cox.

We disagree that this article should be an appendix or extended discussion section to Härtel et al. (2022). This publication dealt with problems related to the uses of age vs. eU plots due to spurious correlation. Hence – they deal with the mathematical reasons of observed correlations that many users of this plot try to interpret in a geological meaningful way. The D-P plot was suggested in these papers as an alternative plot that is mathematically correct. However, none of these papers suggest how to analyze thermochronological data in general or the D-P plot specifically.

Based on this and previous comments, we agree that the goal of this manuscript needs to be better formulated. In addition, we expanded our revised manuscript to include several examples of published real-world data that we analyzed using our proposed guide for the D-P plot (see comment below).

R1: Beyond those two concerns, I question the practical utility of this approach and I see a real potential for misuse and confusion. Generally speaking, it is best to stick to the simplest possible geological interpretations unless there is a convincing **physical** AND statistical basis for doing otherwise. The examples in figure 1 and sections 2.2 and 3.3 should be backed up by both numerous real world datasets demonstrating the pattern and **independent physical evidence** that the purported physical phenomena are the reason that the patterns exist.

As it stands, the manuscript does not contain very much real data, and it makes a number of assertions that are not backed up by any such data.

A: We agree with the suggestion of supporting our conceptual patterns in the D-P plots with actual geological data. We will add several well-studied examples for all D-P relationships shown in Figure 1 to our manuscript including both, fission track and (U-Th)/He datasets.

R1: For example, in line 163-164, the authors assert that "the arithmetic mean age does usually not provide a reliable sample age." The citations provided are again Vermeesch 2008 and Hartel et al. 2022. This argument does not appear to be developed here or in the 2022 paper. The Vermeesch 2008 paper, on the other hand, demonstrates that this is true only in extremely rare special cases.

A: Concerning the arithmetic mean age as sample age, the submitted manuscript and the cited sources support our argument:
(1) Section 3.3 of this manuscript points to Appendix C, which explicitly discusses the bias of the arithmetic mean age.
(2) The last paragraph of section 2 in Härtel et al. (2022) provides a short discussion on the effects of systematic offsets on mean ages. The supplementary material of that article also provides the equations for mean-age bias (arithmetic, geometric and harmonic) caused by random error of D and P.
(3) As Dr. Cox correctly points out, the abstract of Vermeesch (2008) mentions that the difference between the arithmetic mean and the central age is "relatively small" in the case of typical reproducibility of single-grain ages in (U-Th)/He dating. This is not the same as the arithmetic being reliable. In addition, this assumption may not hold for fission-track data, whose reproducibility varies strongly with the track counts.
The main text of that article presents different sample ages for (U-Th)/He dating instead of the arithmetic mean, with each having its specific advantages and applications (section 3). In the case of parentless helium (section 3.2), for example, Vermeesch (2008) recommends using the isochron age; in this case neither the central age nor the arithmetic mean age represent the sample age well.

R1: Figures 4 and 5 provide the only real data, but they do not provide much evidence for the assertions made in figure 1. Figure 4a shows significant statistical scatter, enough that it clearly does not matter exactly how the age is calculated. Figure 4b seems like a better example of the uncertainty being inappropriately reduced (again without a statistically significant change in the reported age) on the basis of a questionable claim about parentless helium than an example of this technique working. And the data in figure 4c are apparently not interpretable regardless of the presentation method.

A: In general, Figure 4 seems to have caused quite some confusion. Its aim was to provide examples of using the workflow in Figure 2, not to provide evidence for the D-P relationships in Figure 1.

In Figure 4a, there is little difference between the pooled and the isochron age not because of statistical scatter, but because D and P are approximately proportional. Sample ages deviate more from each other the more the D-P data differ from proportionality (see Appendix C). We agree that the isochron and the pooled age in Figure 4b overlap within 2s uncertainties. However, this does not mean they are the same, as can be seen from the large difference between the uncertainties and the intercept of the isochron, which is significantly different from zero. The data in Figure 4c are difficult to interpret, but their scatter does not mean they are uninterpretable (see section 3.3.6). The scatter may be due to kinetic differences between the grains due to a property not captured by the measurements, but even without this information it is important to document the D-P relationship (see Table 1), instead of not showing the data.

We will make an effort to re-work Figure 4 and section 3.4 to clarify their purpose and potentially exchange some of the examples to show a more diverse range of scenarios.

R1: Figure 5 shows another pitfall of insisting on plotting D vs P rather than age vs eU because of concern over rare cases of spurious correlations.

A: It is not our intention to insist on analyzing thermochronological data using only the D-P plot (see section 3.3). We prioritize showing D-P plots in Fig. 5 because they are the focus of this manuscript; the original sources of these data display other plots. Our reason to stress the importance of avoiding spurious correlations is their frequent occurrence in ratio-vs.-denominator plots (see textbooks by Chayes, 1978; Rollinson, 1993).

R1: Without age on the plot, the usefulness of these figures in making actual geological interpretations is much reduced compared to the original presentations.

A: We agree that it is often useful to compare single-grain ages to other parameters, e.g. grain size or chemistry. However, our approach prioritizes detecting systematic offsets and analyzing outliers from the D-P relationship (see L.31), before moving on to more specific data-analysis tools (e.g., section 3.3.2, Figure 3c).
We will make sure to state this explicitly in the revised manuscript.

R1: And it does not seem that the authors are asserting that the original sources of these datasets made any mistakes or missed any interpretations on the basis of how they chose to present the data.

A: The fact that we do agree with the original interpretation of these selected examples, does not mean that our approach does not work. The purpose of Figure 5 was to show the D-P plot applied to multi-sample datasets, e.g. for pooling small datasets and comparing them to modelling results (5a), comparing data with to external parameters such as spatial information (5b), or directly comparing the daughter retention of different minerals in the same sample (5c). We understand that it did not reach this goal, and therefore decided to remove it from the manuscript.

R1: If the purpose of this paper is ostensibly to provide a practical "workflow" for real world samples, it should present real world examples showing a significant benefit from an approach that will otherwise add complication, confusion, and the potential for overinterpretation to current workflows.

A: We agree that showing D-P plots from real datasets displaying relationships in Figure 1 would be a strong addition to the manuscript.
We will therefore include new figures with a range of D-P patterns of published datasets and add a comparison of the D-P and its properties compared to other plots (e.g. age-eU, radial or age-grainsize plots).

R1: If it is not the case that these scenarios are commonplace and can be readily distinguished, the proposed data reduction workflow is at best unnecessary and at worst will lead to spurious overinterpretations of scattered or otherwise flawed datasets.

A: We agree with Dr. Cox that for implementing a new data-analysis scheme, it is advantageous to show that it is more than a theoretical concept. In our opinion, the proposed analytical strategy is relatively robust to over-interpretation because the D-P plot displays the data without a preconceived idea regarding the cause of age variation. Relying on a simple scatter plot with two independent axes representing minimally processed data facilitates recognizing non-ideal (not necessarily 'flawed') datasets.
We will add well-documented examples to the manuscript to support this point.

**Referee #2 (anonymous):**

We are pleased to receive the comments of reviewer #2 as they stimulate an in-depth discussion of data analysis. While the reviewer approves our general approach, there are some questions about the usefulness of the D-P plot. Essentially, the reviewer raises the concern that the D-P plot might not be necessary for analyzing data due to the multitude of readily available tools. In response to the reviewer's concerns, we will emphasize the unique benefits of the D-P plot in the introducing sections of our revised manuscript and clarify our motivation for providing this workflow.

Below we answer each of the comments; R2 marks the reviewer#2 comments and A the authors' replies.

R2: The manuscript "The daughter-parent plot: a tool for analyzing thermochronological data" by Härtel and Enkelmann presents the case for using a "daughter-parent" plot for thermochronological data. The manuscript is well written and follows a logical progression of introduction of the technique, background, and application of the method using synthetic and real examples. The main idea of the paper is the presentation of the D-P plot and assigning a decision-based classification scheme based on data relationships.

My initial concern is that the paper is reframing established geochronological concepts that may further confuse readers. Therefore, my concerns may be more related to the specific framing of the paper. Their manuscript shows various synthetic and some real examples but in my opinion does not present a strong argument for adoption of the "D-P plot." I don't understand why the D-P plot should be used in place of any of the other well established characterization methods that are discussed in the paper (e.g., isochron).

A: We are glad to see the general positive response of the reviewer, but see a misunderstanding that occured. Our main argument is not that the current data-analysis schemes are wrong, but that there are unique benefits of using the D-P plot (see below), that are underappreciated because of few thermochronological studies actually applying it. Besides, we envision the D-P plot as a simple device to navigate through the large number of data-analysis tools available, some of which are applicable to a given sample/dataset while others are not. Our aim is therefore to point out the benefits of D-P plotting for thermochronological data and provide guidance for integrating the D-P plot into a data-analysis scheme.

We will clarify our motivations in the introduction of the revised manuscript.

R2: There also seems to be considerable material overlap with previous paper(s) by the first author, specifically their 2022 paper (EPSL v. 599).

A: We will reduce the overlap with our previous works in the revised manuscript.

R2: To reiterate, a detail I keep coming back to as I read the manuscript is that in most cases, the pooled/central age and isochron ages etc. seem to be the end goal of the decision-tree approach—but those methods already have established and definable criteria for their use and have been used in the geochronology community for decades. They are conveniently summarized in the Figure 2 text in the manuscript and Section 3.3. Why should the D-P plot be used with or in place of the pre-existing methods, especially if it is an intermediate step? The D-P plot seems to be a reinvention (or redefinition) of criteria that are already utilized for determining the model age of a thermochronological sample.

A: We agree that there are established criteria for using certain sample ages or data-analysis tools and we do not wish to contest these. The reviewer is also correct that the D-P plot contains information also provided by other data-analysis tools. Still, we argue that it is not only a redefinition of known criteria, but actually has its own unique benefits:
(1) it is the only thermochronological data plot for detecting systematic offsets in daughter or parent concentrations (e.g. Vermeesch, 2008 for "parentless He" from inclusions in apatite), which compromise most data-analysis tools (section 3.3.3).
(2) (2) It enables us to detect radiation-damage effects on daughter retention without being misled by spurious correlation (e.g. Carter, 1990; Härtel et al., 2022).
(3) (3) It allows identifying outliers in the D-P plot with respect to their daughter and parent concentrations and not the single-grain age alone (e.g., He et al., 2021). Examining datasets for outliers is essential for data analysis, and the D-P plot provides an additional perspective.
We will emphasize these benefits in section 2 of the revised manuscript and move the description of the impact of systematic offsets from section 3.3.3 to this section.

R2: For example, when assessing a dataset of an assumed single population of grain ages, the choice between pooled age versus central age is mostly a statistical one.

A: The reviewer correctly points out that the choice between the pooled and central age usually follows a statistical criterion, for example the $\chi^2$ test. For this decision no D-P plot is necessary, however, the D-P plot can be a criterion to assume a single age population beforehand, or, more importantly, point at a systematic offset or radiation-damage-dependent retention. Both of these factors may (1) cause the $\chi^2$ test to fail, and (2) compromise an interpretation based on the pooled or central age.

R2: We already use radial plots and KDEs for identifying and quantifying different age populations, why should the D-P plot also be used? ...or used instead of these other plots?

A: Again, we agree with the reviewer that there is nothing wrong with using KDE or radial plots; these are the tools of choice to analyze different age populations. However, we argue that there are some considerations necessary before applying them, to make sure that shoulders in the KDE curve are not due to statistical outliers or systematic offsets; or that

observed patterns in the radial plot reflect age populations rather than age-uncertainty correlations (e.g., Jonckheere et al., 2024). Assessing the age variation, finding outliers (section 3.3), or testing for systematic offsets (section 3.3.3) can be done conveniently by the D-P plot. On top of that, the D-P plot may already point to distinct age populations in a sample. That is why we suggest the D-P plot as first step for the data analysis and suggest the KDE and radial plots as follow-up tools if different age populations seem plausible.

R2: Another point is that if data display some weak trend but have large uncertainties, they of course can be treated as a "cluster"—this happens quite frequently with fission-track datasets generated by the external detector method. Again, why should the D-P plot be used instead of, or in addition to, pre-existing methods?

A: Again, there is nothing wrong with using the classical methods instead of the D-P plot to analyze clustered data, if one can exclude systematic effects. Still, the D-P plot may be a convenient visualization tool to reach the conclusion that such data form a cluster as opposed to a linear relationship. In addition, the D-P plot shows how much of the individual age uncertainty stems from induced- and spontaneous-track counts; this is especially useful for comparing samples with different U content at a glance.

R2: Some of the arguments posed do not always hold up. EDM data characterized by large single-grain uncertainties may actually mask real age populations and therefore the data should NOT be treated as a data cluster. Would the D-P plot be useful in such a case? In some instances, other data (apatite composition, U-Pb ages, provenance, etc.) tell us there are likely multiple kinetic populations. This must be assessed by a skilled and experienced thermochronologist. As we well know, statistical methods are not infallible. The data are the data, and usually do not conform to our expectations.

A: We thank the reviewer for bringing up this example. We agree that in this special case the interpretation as a cluster is not necessarily correct (see section 3.3.2), but that the careful comparison to other, independent data is necessary to separate the different populations. Color-coding data in the D-P plot as shown in Figure 3c helps to test for relationships between age and grain properties, but also to discern possible relationships of these properties (e.g., Cl content in apatite) to the parent or daughter concentrations.
We will add this argument to section 3.3.2 in the revised manuscript.

R2: This is my opinion, but I generally think that strict decision-based approaches are problematic for some types of data analysis, mainly because they are often subjective in the sense that they are based on a few known or conjured scenarios. That is not to say such approaches are entirely without merit, but it can be a slippery slope in terms of their careful use and application. What happens when data do not adhere to the provided scheme?

A: It is correct that the weak spot of a strict decision-based approach is that some datasets may not be assignable unambiguously to a class. That is why our sections 3.2 and 3.3 provide additional discussion to the criteria used in the workflow (Figure 2), point out alternative classifications, e.g., in the complicated case of scattered data, and suggests further tools or gathering constraints from other data.

On the other hand, we think that a decision-based data-analysis scheme is helpful to keep the analysis process consistent, especially for studies involving many samples. On top of that, decisions based on simple criteria also enable inexperienced analysts to choose the right plots and interpretation tools rather than randomly applying them.

R2: I vaguely understand the intention behind the D-P plot, but in reality most users simply (and rather unfortunately) want The Answer and don't want to spend lots of time doing detailed data analysis and interpretation. This is problematic and thus presents an opportunity for blind application by users, which is a nontrivial matter in thermochronological data interpretation. An analogous case is the practice of binning and averaging U-Th/He dates by eU for modeling—there is simply no legitimate or statistical rationale for doing it.

A: We agree that proper data analysis is not very attractive and often overlooked by many users, translating their data too quickly into geological models. In our opinion, this is also a function of the availability of easy-to-use software tools and the accessibility of data analysis literature for carrying out the analysis by users that get data delivered from laboratories.
That is why we will add a new section on Incaplot (Härtel, 2023) to the revised manuscript. Incaplot is a free graphical-user-interface program dedicated to producing D-P plots for thermochronological data in a straightforward manner with options for isochron fitting, pooled- and single-grain age calculation. The program is already in use in the Calgary Geo- and Thermochronology group and further updates will enhance its capabilities and usefulness.

R2: Outside of mineral age standards, how often do thermochronology datasets not meet the "small dataset" consideration in Section 4.1? The authors essentially define this case as the norm and say the D-P plot can't really be used other than as a "qualitative guide". This is a bit confusing.

A: We state in section 4.1 that this problem mainly concerns whole-grain (U-Th)/He data where 5 or less single-grain ages are common. For other methods such as FT dating, or laser-ablation based (U-Th)/He dating or zircon Raman dating, this is not a concern because >15 ages are typically measured per sample. Small samples are not only a problem for the D-P plot, but for any plots used in thermochronology to identify patterns/trends (age vs. grain size, age vs eU, radial plot). This is because of the data potentially not representing the sample well, hence, the question is if observed trends are actually significant.

R2: Figure 5 presents a good example of showing data in He (nmol/g; daughter) vs. eU (ug/g; parent)—how do those plots compare to the normal plots of "apparent age" vs. eU?

That should be shown. If they are the different, is there an advantage of the D-P plot? If they are the same, why use it?

A: Yes, we agree that comparing the D-P plot with the age-eU plot is useful. However Härtel et al., 2022 specifically dealt with spurious correlation in the age-eU plot that does not affect the D-P plot. Still, we will include age-eU plots in the revised manuscript, pointing at this problem.

R2: It is not clear to me why the datasets are shown in Figure 5? What is the message of this figure? For example, fig. 5a shows Miltich (2005) zircon helium data with different radiation damage relationship prediction curves for different thermal history scenarios. Is there a damage scenario the authors prefer or are they arbitrarily shown, and if arbitrary, why show them at all? Are the high-He grains outliers or is there just data scatter? Fig. 5c shows eU and He concentration for 3 different minerals. Suffice to say, they should plot differently since each mineral has different and characteristic parent/daughter isotopes, depending upon the mineral. What should the takeaway of this figure be?

A: We appreciate the reviewer's observations on Figure 5. Our intention was to show possible uses of D-P plots for comparing multiple samples from the same geological context or different thermochronometers from the same. It seems like this figure caused more confusion than it helped to understand the concept. That is why we will remove it from the manuscript. Following the request of reviewer #1, we will add real-world data to the manuscript and hope that this will be more useful to the overall concept of the paper.

R2: Line 17: what is meant by "consistent and traceable" with respect to thermochronometric data analysis? When someone plots U-Th/He dates versus eU, are there obvious issues with consistency and how it was done?

A: We mean that by reporting a type of D-P relationship instead of only a sample age or t-T model, it is easier for other thermochronologists to understand the data-analysis process. This is less on how someone did a certain plot, but on tracing why they chose to do so.
We will revise this section of the manuscript for clarification.

R2: Line 21: use of word "ingredient" is odd, maybe change that.

A: We will revise this sentence.

R2: Figure 4c: remove the dashed lines and min/max ages. As shown this looks like a radial plot and is misleading.

A: As the reviewer correctly points out, displaying an age as a line in the D-P plot works in the same way as in the radial plot, but we see no risk of mixing up the two due to the different plot construction and labels. We will therefore keep the lines and ages in this figure.

---

## Author Response (AR3)

**Manuscript gchron-2024-1 - Revision 2**

**The daughter-parent plot: a tool for analyzing thermochronological data**

Birk Härtel and Eva Enkelmann

———————————————————————————————————————————————

Dear Editor,

We are pleased to receive the new comments by referee #2. The reviewer appreciates the improvements we have made in response to previous comments and contests neither the general usefulness of the D-P plot nor our practical advice for data analysis. We accept most of the reviewer's suggestions, which focus on the presentation of our data examples.

In response to these suggestions we modified our manuscript as follows: we reorganized the example figures and text in sections 2.2 and 2.3 to better separate the description of different data patterns. We also tried to better delineate the difference between the theoretical background in section 2 and the practical considerations in section 3.

In our opinion, these modifications enhance the readability of our manuscript, especially for readers inexperienced in thermochronological data analysis. We hope that the revised manuscript can be considered for publication in *Geochronology*. Please find below the replies to the reviewer comments.

Kind regards,

Birk Härtel for the co-authors

**Referee #2 (anonymous)**

We appreciate the suggestions by referee #2 concerning the presentation of our data examples and the need to better justify the use of the D-P plot.

Below we respond to each of these comments in detail. **R2** marks the reviewer comments and **A** the authors' replies.

**R2:** The revised manuscript by Härtel and Enkelmann is improved from the original submission, and I suggest minor revisions mostly regarding organization and figures. The introduction on the background, strengths, and specific differences between traditional isochron plots makes the apparent utility of D-P plots much more obvious than before (e.g., no error correlation between variables as in isochron plots). The paper still seems to tread on topics laid out in a previous publication and by their own admission represents a gray intermediary between traditionally utilized data plots and more detailed data interpretation methods. I think this point needs more development.

**A:** We understand that more justification for using the D-P plot over, e.g, the age-eU plot would make the manuscript more attractive. Some of these arguments are laid out in detail in Härtel et al. (2022). We added Figure 10 as an example to section 2.3, and provided a reference to the more detailed work on spurious associations in Härtel et al. (2022).

**R2:** The broader utility of the D-P plot remains elusive in my estimation, but I can see how it may be useful in very specific cases. The text does a better job of explaining utility (compared to the initial submission), but the figures do not support this improvement as clearly or efficiently. The figures should stand alone from the text, and in a paper specifically about plotting, the plots in the figures could better reflect the messaging of the paper.

The addition of D-P plots of previously published datasets from the literature is somewhat helpful in this regard but falls short due to the figures not distinctly showing the benefits of the D-P visualization. The figures often combine many different datasets and "trend styles" making the text and figure content hard to follow throughout (e.g., Sections 2.2 and 2.3 jump around a lot). This could all be simplified and thus the text made clearer in the process. If the focus of the paper is to justify usage of the D-P plot and advertise the advantages, then the figures should clearly reflect that goal. I recommend that the paper focus on a few very clear examples. This could alternatively be achieved by subheadings specifically breaking out each example from figure 1. This may sound tedious, but it makes it much more useful for the reader if Section X and Figure X are discussing one/the same topic.

**A:** We appreciate this criticism on our presentation of the data examples. We agree that the figures should be able to stand on their own and that the connection between the description in section 2.2 and each of the D-P relationships should be clearer. We therefore reduced the number of examples and split the figures by plotting the example for each relationship together with the conceptual plots from Figure 1, and associated each with a new subheading in section 2.2.

Concerning the utility of the plot, we split the previous Figure 5 to better reflect the arguments given in section 2.3. The two new figures now show: (1) the impact of systematic offset on the radial plot, the $\chi^2$ statistic, and the central age, and (2) how the D-P plot is an improvement on the age-eU plot for detecting radiation-damage.

**R2:** For example, the D-P plot seems to be useful for identifying linear trends with either zero or non-zero intercepts, but WHY is that advantageous? The text goes into this a little, but the figures could better reflect the advantages.

**A:** We improved on the figures and text in section 2.3 to clarify to illustrate how overlooking systematic offsets compromises data interpretation.

**R2:** In Figure 2a, what is the accepted FCT age and how does the manner of plotting/pooled age calculation provide insight and/or better agreement with that age?

**A:** We added the reference age of FCT to the manuscript for comparison. In section 2.3 an in the conclusions we also emphasized the importance of recognizing the D-P relationship before choosing other data-analysis tools.

**R2:** Another recommendation would be to show very clear examples of when traditional date-eU or date-kinetic parameter plots diverge in behavior that can be explained or turned into some useful insight based on the D-P relationship. If the data are just being recast or undergoing a transformation just to be in D-P space for no other reason than the authors preference, then the reader will be unlikely to engage with the approach further.

**A:** We added a reference to Härtel et al. (2022) and a figure (Fig. 10) to section 2.3, showing that the age-eU plot displays an association for a true radiation-damage effect (True Positive), but also for scattered or systematically offset data (False Positive). In contrast, the D-P plot shows distinguishable patterns for each of these cases.

**R2:** For example, I expected this in Figure 5 with the Orme et al. data in panel A (confusing as presented). What should be taken away from panel A?

**A:** We reorganized Figure 5 (now Figure 9) to show (a) a D-P, (b) a radial, and (c) an age-eU plot to show how the systematic offset is only visible in (a) but leads to significant bias in (b) and a spurious (positive) association in (c).

**R2:** I would highly suggest combining figure 1 and the various other figures together to show one figure with, for instance, on the left side "hypothetical example" and right column "real example." This may be a better approach that would avoid jumping between early and late figure references. For example, a hypothetical "inverse" trend from Figure 1h and the Figure 4c Miltich data as the real example combined into one larger figure with clearly labelled columns/panels may be a better option? If the hypothetical and literature examples were combined in a single figure, then the text could easily discuss each in turn and reference the figure accordingly. This may help with organization and flow. Later figures may also delve into specific aspects of, say, details about why the D-P plot for published dataset X shows some pattern and how it could be interpreted.

**A:** We thank the reviewer for this useful suggestion. We applied it in the revised manuscript by pairing each of the relationships of the former Figure 1 with its respective counterpart(s) from Figs. 2-4.

**Other comments:**
**R2:** One phrase at line 78 is a bit confusing to me: "Summarizing thermochronological ages by a mean age without examining the daughter-parent relationship thus does injustice to the data and may neglect important information." What information is being wholly neglected outright (?) and we have already turned away from using mean ages for most thermochronological mineral systems where dates have a direct dependence or relationship with some kinetic variable.
Traditional plots utilizing eU (for U-Th/He), or elemental/compositional information in the case of AFT, are often examined in a similar manner with respect to "date(s)" but are not focused on the mean age—usually quite the opposite. For both zircon and apatite He data, mean ages are essentially useless due to radiation damage effects—a central tendency estimate for AFT data (either arithmetic mean or central age) is useful only when single-grain ages are kinetically homogenous and/or pass the chi-squared test (the latter being

somewhat problematic with high-n/high-precision single-grain data encountered with the ICPMS method). Figure 1 shows various data trend examples of similar "mean age" but again, all of these examples—clusters, non-linearity, multiple populations, scattered data, and inverse trends would all also be displayed in a date-eU or age-kinetic parameter plot for He or AFT data. This again comes back to a critique of the initially submitted manuscript.

**A:** We agree that the notion of a sample age becomes less and less important with our increasing understanding of thermochronological dating system, giving this statement an unclear direction. We meant to point out at the problem of deciding on an analysis tool (e.g., sample age, radial plot, isochron, age-eU, $\chi^2$) before checking if such a tool is meaningful for a given data pattern (see section 2.3 on problems with systematic offsets). We decided to better clarify this point in sections 2.2 and 2.3 and deleted this phrase.

**R2:** In Figure 2a—is the "pooled age" unique to using the D-P information or is that just a pooled age is the classic sense using the single-grain data? I suspect the former based on the caption, but I see this as a useful metric and support for D-P plotting with an analytical age standard if specified.
A D-P plot with pooled age of 850 Ma carries what meaning?

**A:** Our pooled age is the classic pooled age as explained in Appendix C and thus has the same meaning (e.g., Galbraith (2005)). Plotting the pooled age as a line in the D-P plot allows us to evaluate if the single-grain data are in agreement with the pooled age as sample age, and if there is significant difference between a pooled and an isochron age (see section 3.4.1). We added these considerations to section 3.4.2.

**R2:** If multiple "trend types" are going to be shown in one figure (I'd advise against this), then they should be clearly labelled in the figure as "linear," "cluster", etc. Why is the data 'cluster' in figure 2b shown with the linear examples and what in the value of the D-P plot for such data?

**A:** We thank the reviewer for pointing out this confusion. In response, we split up the shown examples in section 2.2. The D-P plot for the clustered data is mainly a tool to identify the data as a cluster.

**R2:** Overall, the text in Section 2.2 seems rushed.

**A:** We reorganized section 2.2 with subheadings, individual figures, and a revised text to make it more readable.

**R2:** Figure 3ab zircon He examples again display a similar radiation damage relationship as observed in a date-eU plot. What do the date-eU plots look like? Are they similar?

**A:** They show an association in age-eU, but it is not distinguishable from spurious correlation. We added a figure (Fig. 10) to compare the differences between radiation-damage effects and other sources of age variation between the D-P and age-eU plot.

**R2:** Figure 3c nicely shows a multikinetic AFT relationship with two populations from Issler et al. 2005, but again this would also be apparent in an age vs. Cl plot as well...or would it? A direct comparison between such plots is necessary to justify usage of the D-P plot, at least to align with ideas presented in the text.

**A:** We agree that the age-Cl and the D-P plot should give the same result in such a case. We therefore do not suggest to use the D-P plot exclusively, but suggest its use together with the more traditional tools (section 3.4, Fig. 11). The D-P plot, however, is necessary as a first step (see section 2.3), because the single-grain ages or other plots could be biased. Starting out with an age-Cl or age-eU plot may therefore be misleading. In contrast, using the D-P plot first avoids these problems, and helps us to choose more specific plots (e.g., age-Cl) for subsequent data analysis. We added this explanation to section 2.3 together with a revised version of Figure 5 (Figs. 10 and 11) pointing out biases in some traditional data-analysis tools.

**R2:** Figure 4—wouldn't these data be scattered in a date-eU or MWAR-eU plot as well?

**A:** We added a figure to section 2.3 (Fig. 10) showing that the age-eU trend can be fairly systematic given a scattered D-P relationship, even if the age variation is due to different grain sizes (Fig. 10f). Again, we do not suggest to use the D-P plot exclusively, but warn against analyzing data without examining the D-P relationship.

**R2:** Section 3 seems out of place and some parts of it may not be necessary in the main text, as they distract a bit from the flow.

**A:** From our perspective, section 3 is the core of this article, providing practical guidance on how to analyze data in D-P space. We added an explicit statemtn to the introduction to clarify the contrast between section 3 and the more theoretical section 2.